# Efficient Second Order Online Learning by Sketching

**Haipeng Luo**
Princeton University, Princeton, NJ USA
haipengl@cs.princeton.edu

**Alekh Agarwal**
Microsoft Research, New York, NY USA
alekha@microsoft.com

**Nicolò Cesa-Bianchi**
Università degli Studi di Milano, Italy
nicolo.cesa-bianchi@unimi.it

**John Langford**
Microsoft Research, New York, NY USA
jcl@microsoft.com

## Abstract

We propose Sketched Online Newton (SON), an online second order learning algorithm that enjoys substantially improved regret guarantees for ill-conditioned data. SON is an enhanced version of the Online Newton Step, which, via sketching techniques enjoys a running time linear in the dimension and sketch size. We further develop sparse forms of the sketching methods (such as Oja's rule), making the computation linear in the sparsity of features. Together, the algorithm eliminates all computational obstacles in previous second order online learning approaches.

## 1 Introduction

Online learning methods are highly successful at rapidly reducing the test error on large, high-dimensional datasets. First order methods are particularly attractive in such problems as they typically enjoy computational complexity linear in the input size. However, the convergence of these methods crucially depends on the geometry of the data; for instance, running the same algorithm on a rotated set of examples can return vastly inferior results. See Fig. 1 for an illustration.

Second order algorithms such as Online Newton Step [18] have the attractive property of being invariant to linear transformations of the data, but typically require space and update time quadratic in the number of dimensions. Furthermore, the dependence on dimension is not improved even if the examples are sparse. These issues lead to the key question in our work: *Can we develop (approximately) second order online learning algorithms with efficient updates?* We show that the answer is "yes" by developing efficient sketched second order methods with regret guarantees. Specifically, the three main contributions of this work are:

**1. Invariant learning setting and optimal algorithms (Section 2).** The typical online regret minimization setting evaluates against a benchmark that is bounded in some fixed norm (such as the $\ell_2$-norm), implicitly putting the problem in a nice geometry. However, if all the features are scaled down, it is desirable to compare with accordingly larger weights, which is precluded by an apriori fixed norm bound. We study an invariant learning setting similar to the paper [33] which compares the learner to a benchmark only constrained to generate bounded predictions on the sequence of examples. We show that a variant of the Online Newton Step [18], while quadratic in computation, stays regret-optimal with a nearly matching lower bound in this more general setting.

**2. Improved efficiency via sketching (Section 3).** To overcome the quadratic running time, we next develop sketched variants of the Newton update, approximating the second order information using a small number of carefully chosen directions, called a *sketch*. While the idea of data sketching is widely studied [36], as far as we know our work is the first one to apply it to a general adversarial

online learning setting and provide rigorous regret guarantees. Three different sketching methods are considered: Random Projections [1, 19], Frequent Directions [12, 23], and Oja's algorithm [28, 29], all of which allow linear running time per round. For the first two methods, we prove regret bounds similar to the full second order update whenever the sketch-size is large enough. Our analysis makes it easy to plug in other sketching and online PCA methods (e.g. [11]).

**3. Sparse updates (Section 4).** For practical implementation, we further develop sparse versions of these updates with a running time linear in the sparsity of the examples. The main challenge here is that even if examples are sparse, the sketch matrix still quickly becomes dense. These are the first known sparse implementations of the Frequent Directions[1] and Oja's algorithm, and require new sparse eigen computation routines that may be of independent interest.

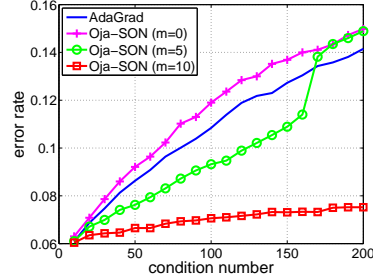

Figure 1: Error rate of SON using Oja's sketch, and ADAGRAD on a synthetic ill-conditioned problem. $m$ is the sketch size ($m = 0$ is Online Gradient, $m = d$ resembles Online Newton). SON is nearly invariant to condition number for $m = 10$.

Empirically, we evaluate our algorithm using the sparse Oja sketch (called Oja-SON) against first order methods such as diagonalized ADA-GRAD [6, 25] on both ill-conditioned synthetic and a suite of real-world datasets. As Fig. 1 shows for a synthetic problem, we observe substantial performance gains as data conditioning worsens. On the real-world datasets, we find improvements in some instances, while observing no substantial second-order signal in the others.

**Related work** Our online learning setting is closest to the one proposed in [33], which studies scale-invariant algorithms, a special case of the invariance property considered here (see also [31, Section 5]). Computational efficiency, a main concern in this work, is not a problem there since each coordinate is scaled independently. Orabona and Pál [30] study unrelated notions of invariance. Gao et al. [9] study a specific randomized sketching method for a special online learning setting.

The L-BFGS algorithm [24] has recently been studied in the stochastic setting[2] [3, 26, 27, 34, 35], but has strong assumptions with pessimistic rates in theory and reliance on the use of large mini-batches empirically. Recent works [7, 15, 14, 32] employ sketching in stochastic optimization, but do not provide sparse implementations or extend in an obvious manner to the online setting. The Frank-Wolfe algorithm [8, 20] is also invariant to linear transformations, but with worse regret bounds [17] without further assumptions and modifications [10].

**Notation** Vectors are represented by bold letters (e.g., $\boldsymbol{x}$, $\boldsymbol{w}$, ...) and matrices by capital letters (e.g., $M$, $A$, ...). $M_{i,j}$ denotes the $(i, j)$ entry of matrix $M$. $\boldsymbol{I}_d$ represents the $d \times d$ identity matrix, $\boldsymbol{0}_{m \times d}$ represents the $m \times d$ matrix of zeroes, and $\text{diag}\{\boldsymbol{x}\}$ represents a diagonal matrix with $\boldsymbol{x}$ on the diagonal. $\lambda_i(A)$ denotes the $i$-th largest eigenvalue of $A$, $\|\boldsymbol{w}\|_A$ denotes $\sqrt{\boldsymbol{w}^\top A \boldsymbol{w}}$, $|A|$ is the determinant of $A$, $\text{TR}(A)$ is the trace of $A$, $\langle A, B \rangle$ denotes $\sum_{i,j} A_{i,j} B_{i,j}$, and $A \preceq B$ means that $B - A$ is positive semidefinite. The sign function $\text{SGN}(a)$ is 1 if $a \geq 0$ and $-1$ otherwise.

## 2 Setup and an Optimal Algorithm

We consider the following setting. On each round $t = 1, 2 \ldots, T$: **(1)** the adversary first presents an example $\boldsymbol{x}_t \in \mathbb{R}^d$, **(2)** the learner chooses $\boldsymbol{w}_t \in \mathbb{R}^d$ and predicts $\boldsymbol{w}_t^\top \boldsymbol{x}_t$, **(3)** the adversary reveals a loss function $f_t(\boldsymbol{w}) = \ell_t(\boldsymbol{w}^\top \boldsymbol{x}_t)$ for some convex, differentiable $\ell_t : \mathbb{R} \to \mathbb{R}_+$, and **(4)** the learner suffers loss $f_t(\boldsymbol{w}_t)$ for this round.

The learner's regret to a comparator $\boldsymbol{w}$ is defined as $R_T(\boldsymbol{w}) = \sum_{t=1}^T f_t(\boldsymbol{w}_t) - \sum_{t=1}^T f_t(\boldsymbol{w})$. Typical results study $R_T(\boldsymbol{w})$ against all $\boldsymbol{w}$ with a bounded norm in some geometry. For an invariant update,

we relax this requirement and only put bounds on the predictions $\boldsymbol{w}^\top \boldsymbol{x}_t$. Specifically, for some pre-chosen constant $C$ we define $\mathcal{K}_t \stackrel{\text{def}}{=} \{\boldsymbol{w} : |\boldsymbol{w}^\top \boldsymbol{x}_t| \le C\}$. We seek to minimize regret to all comparators that generate bounded predictions on every data point, that is:

$$R_T = \sup_{\boldsymbol{w} \in \mathcal{K}} R_T(\boldsymbol{w}) \quad \text{where} \quad \mathcal{K} \stackrel{\text{def}}{=} \bigcap_{t=1}^{T} \mathcal{K}_t = \{\boldsymbol{w} : \forall t = 1, 2, \ldots T, \ |\boldsymbol{w}^\top \boldsymbol{x}_t| \le C\} \ .$$

Under this setup, if the data are transformed to $M\boldsymbol{x}_t$ for all $t$ and some invertible matrix $M \in \mathbb{R}^{d \times d}$, the optimal $\boldsymbol{w}^*$ simply moves to $(M^{-1})^\top \boldsymbol{w}^*$, which still has bounded predictions but might have significantly larger norm. This relaxation is similar to the comparator set considered in [33].

We make two structural assumptions on the loss functions.

**Assumption 1.** *(Scalar Lipschitz) The loss function $\ell_t$ satisfies $|\ell'_t(z)| \le L$ whenever $|z| \le C$.*

**Assumption 2.** *(Curvature) There exists $\sigma_t \ge 0$ such that for all $\boldsymbol{u}, \boldsymbol{w} \in \mathcal{K}$, $f_t(\boldsymbol{w})$ is lower bounded by $f_t(\boldsymbol{u}) + \nabla f_t(\boldsymbol{u})^\top (\boldsymbol{w} - \boldsymbol{u}) + \frac{\sigma_t}{2} \left( \nabla f_t(\boldsymbol{u})^\top (\boldsymbol{u} - \boldsymbol{w}) \right)^2$ .*

Note that when $\sigma_t = 0$, Assumption 2 merely imposes convexity. More generally, it is satisfied by squared loss $f_t(\boldsymbol{w}) = (\boldsymbol{w}^\top \boldsymbol{x}_t - y_t)^2$ with $\sigma_t = \frac{1}{8C^2}$ whenever $|\boldsymbol{w}^\top \boldsymbol{x}_t|$ and $|y_t|$ are bounded by $C$, as well as for all exp-concave functions (see [18, Lemma 3]).

Enlarging the comparator set might result in worse regret. We next show matching upper and lower bounds qualitatively similar to the standard setting, but with an extra unavoidable $\sqrt{d}$ factor. [3]

**Theorem 1.** *For any online algorithm generating $\boldsymbol{w}_t \in \mathbb{R}^d$ and all $T \ge d$, there exists a sequence of $T$ examples $\boldsymbol{x}_t \in \mathbb{R}^d$ and loss functions $\ell_t$ satisfying Assumptions 1 and 2 (with $\sigma_t = 0$) such that the regret $R_T$ is at least $CL\sqrt{dT/2}$.*

We now give an algorithm that matches the lower bound up to logarithmic constants in the worst case but enjoys much smaller regret when $\sigma_t \ne 0$. At round $t+1$ with some invertible matrix $A_t$ specified later and gradient $\boldsymbol{g}_t = \nabla f_t(\boldsymbol{w}_t)$, the algorithm performs the following update *before* making the prediction on the example $\boldsymbol{x}_{t+1}$:

$$\boldsymbol{u}_{t+1} = \boldsymbol{w}_t - A_t^{-1} \boldsymbol{g}_t, \quad \text{and} \quad \boldsymbol{w}_{t+1} = \operatorname*{argmin}_{\boldsymbol{w} \in \mathcal{K}_{t+1}} \|\boldsymbol{w} - \boldsymbol{u}_{t+1}\|_{A_t} \ . \tag{1}$$

The projection onto the set $\mathcal{K}_{t+1}$ differs from typical norm-based projections as it only enforces boundedness on $\boldsymbol{x}_{t+1}$ at round $t+1$. Moreover, this projection step can be performed in closed form.

**Lemma 1.** *For any $\boldsymbol{x} \ne \boldsymbol{0}, \boldsymbol{u} \in \mathbb{R}^d$ and positive definite matrix $A \in \mathbb{R}^{d \times d}$, we have*

$$\operatorname*{argmin}_{\boldsymbol{w} \,:\, |\boldsymbol{w}^\top \boldsymbol{x}| \le C} \|\boldsymbol{w} - \boldsymbol{u}\|_A = \boldsymbol{u} - \frac{\tau_C(\boldsymbol{u}^\top \boldsymbol{x})}{\boldsymbol{x}^\top A^{-1} \boldsymbol{x}} A^{-1} \boldsymbol{x}, \quad \text{where } \tau_C(y) = \text{SGN}(y) \max\{|y| - C, 0\}.$$

If $A_t$ is a diagonal matrix, updates similar to those of Ross et al. [33] are recovered. We study a choice of $A_t$ that is similar to the Online Newton Step (ONS) [18] (though with different projections):

$$A_t = \alpha \boldsymbol{I}_d + \sum_{s=1}^{t} (\sigma_s + \eta_s) \boldsymbol{g}_s \boldsymbol{g}_s^\top \tag{2}$$

for some parameters $\alpha > 0$ and $\eta_t \ge 0$. The regret guarantee of this algorithm is shown below:

**Theorem 2.** *Under Assumptions 1 and 2, suppose that $\sigma_t \ge \sigma \ge 0$ for all $t$, and $\eta_t$ is non-increasing. Then using the matrices (2) in the updates (1) yields for all $\boldsymbol{w} \in \mathcal{K}$,*

$$R_T(\boldsymbol{w}) \le \frac{\alpha}{2} \|\boldsymbol{w}\|_2^2 + 2(CL)^2 \sum_{t=1}^{T} \eta_t + \frac{d}{2(\sigma + \eta_T)} \ln \left( 1 + \frac{(\sigma + \eta_T) \sum_{t=1}^{T} \|\boldsymbol{g}_t\|_2^2}{d\alpha} \right) \ .$$

**Algorithm 1** Sketched Online Newton (SON)

---
**Input:** Parameters $C$, $\alpha$ and $m$.
 1: Initialize $\boldsymbol{u}_1 = \boldsymbol{0}_{d \times 1}$.
 2: Initialize sketch $(S, H) \leftarrow \textbf{SketchInit}(\alpha, m)$.
 3: **for** $t = 1$ **to** $T$ **do**
 4:     Receive example $\boldsymbol{x}_t$.
 5:     **Projection step:** compute $\widehat{\boldsymbol{x}} = S\boldsymbol{x}_t$, $\gamma = \frac{\tau_C(\boldsymbol{u}_t^\top \boldsymbol{x}_t)}{\boldsymbol{x}_t^\top \boldsymbol{x}_t - \widehat{\boldsymbol{x}}^\top H \widehat{\boldsymbol{x}}}$ and set $\boldsymbol{w}_t = \boldsymbol{u}_t - \gamma(\boldsymbol{x}_t - S^\top H \widehat{\boldsymbol{x}})$.
 6:     Predict label $y_t = \boldsymbol{w}_t^\top \boldsymbol{x}_t$ and suffer loss $\ell_t(y_t)$.
 7:     Compute gradient $\boldsymbol{g}_t = \ell_t'(y_t)\boldsymbol{x}_t$ and the *to-sketch vector* $\widehat{\boldsymbol{g}} = \sqrt{\sigma_t + \eta_t}\boldsymbol{g}_t$.
 8:     $(S, H) \leftarrow \textbf{SketchUpdate}(\widehat{\boldsymbol{g}})$.
 9:     **Update weight:** $\boldsymbol{u}_{t+1} = \boldsymbol{w}_t - \frac{1}{\alpha}(\boldsymbol{g}_t - S^\top H S \boldsymbol{g}_t)$.
10: **end for**

---

The dependence on $\|\boldsymbol{w}\|_2^2$ implies that the method is not completely invariant to transformations of the data. This is due to the part $\alpha \boldsymbol{I}_d$ in $A_t$. However, this is not critical since $\alpha$ is fixed and small while the other part of the bound grows to eventually become the dominating term. Moreover, we can even set $\alpha = 0$ and replace the inverse with the Moore-Penrose pseudoinverse to obtain a truly invariant algorithm, as discussed in Appendix D. We use $\alpha > 0$ in the remainder for simplicity.

The implication of this regret bound is the following: in the worst case where $\sigma = 0$, we set $\eta_t = \sqrt{d/C^2 L^2 t}$ and the bound simplifies to

$$R_T(\boldsymbol{w}) \leq \frac{\alpha}{2}\|\boldsymbol{w}\|_2^2 + \frac{CL}{2}\sqrt{Td}\ln\left(1 + \frac{\sum_{t=1}^T \|\boldsymbol{g}_t\|_2^2}{\alpha CL\sqrt{Td}}\right) + 4CL\sqrt{Td}\,,$$

essentially only losing a logarithmic factor compared to the lower bound in Theorem 1. On the other hand, if $\sigma_t \geq \sigma > 0$ for all $t$, then we set $\eta_t = 0$ and the regret simplifies to

$$R_T(\boldsymbol{w}) \leq \frac{\alpha}{2}\|\boldsymbol{w}\|_2^2 + \frac{d}{2\sigma}\ln\left(1 + \frac{\sigma\sum_{t=1}^T \|\boldsymbol{g}_t\|_2^2}{d\alpha}\right)\,, \tag{3}$$

extending the $\mathcal{O}(d\ln T)$ results in [18] to the weaker Assumption 2 and a larger comparator set $\mathcal{K}$.

# 3   Efficiency via Sketching

Our algorithm so far requires $\Omega(d^2)$ time and space just as ONS. In this section we show how to achieve regret guarantees nearly as good as the above bounds, while keeping computation within a constant factor of first order methods.

Let $G_t \in \mathbb{R}^{t \times d}$ be a matrix such that the $t$-th row is $\widehat{\boldsymbol{g}}_t^\top$ where we define $\widehat{\boldsymbol{g}}_t = \sqrt{\sigma_t + \eta_t}\boldsymbol{g}_t$ to be the *to-sketch vector*. Our previous choice of $A_t$ (Eq. (2)) can be written as $\alpha \boldsymbol{I}_d + G_t^\top G_t$. The idea of sketching is to maintain an approximation of $G_t$, denoted by $S_t \in \mathbb{R}^{m \times d}$ where $m \ll d$ is a small constant called the sketch size. If $m$ is chosen so that $S_t^\top S_t$ approximates $G_t^\top G_t$ well, we can redefine $A_t$ as $\alpha \boldsymbol{I}_d + S_t^\top S_t$ for the algorithm.

To see why this admits an efficient algorithm, notice that by the Woodbury formula one has $A_t^{-1} = \frac{1}{\alpha}\left(\boldsymbol{I}_d - S_t^\top(\alpha\boldsymbol{I}_m + S_t S_t^\top)^{-1}S_t\right)$. With the notation $H_t = (\alpha\boldsymbol{I}_m + S_t S_t^\top)^{-1} \in \mathbb{R}^{m \times m}$ and $\gamma_t = \tau_C(\boldsymbol{u}_{t+1}^\top \boldsymbol{x}_{t+1})/(\boldsymbol{x}_{t+1}^\top \boldsymbol{x}_{t+1} - \boldsymbol{x}_{t+1}^\top S_t^\top H_t S_t \boldsymbol{x}_{t+1})$, update (1) becomes:

$$\boldsymbol{u}_{t+1} = \boldsymbol{w}_t - \frac{1}{\alpha}\big(\boldsymbol{g}_t - S_t^\top H_t S_t \boldsymbol{g}_t\big), \quad \text{and} \quad \boldsymbol{w}_{t+1} = \boldsymbol{u}_{t+1} - \gamma_t\big(\boldsymbol{x}_{t+1} - S_t^\top H_t S_t \boldsymbol{x}_{t+1}\big)\,.$$

The operations involving $S_t \boldsymbol{g}_t$ or $S_t \boldsymbol{x}_{t+1}$ require only $\mathcal{O}(md)$ time, while matrix vector products with $H_t$ require only $\mathcal{O}(m^2)$. Altogether, these updates are at most $m$ times more expensive than first order algorithms as long as $S_t$ and $H_t$ can be maintained efficiently. We call this algorithm Sketched Online Newton (SON) and summarize it in Algorithm 1.

We now discuss three sketching techniques to maintain the matrices $S_t$ and $H_t$ efficiently, each requiring $\mathcal{O}(md)$ storage and time linear in $d$.

| **Algorithm 2** FD-Sketch for FD-SON | **Algorithm 3** Oja's Sketch for Oja-SON |
|---|---|
| **Internal State:** $S$ and $H$. | **Internal State:** $t$, $\Lambda$, $V$ and $H$. |
| **SketchInit**$(\alpha, m)$ <br> 1: Set $S = \mathbf{0}_{m \times d}$ and $H = \frac{1}{\alpha}\boldsymbol{I}_m$. <br> 2: Return $(S, H)$. | **SketchInit**$(\alpha, m)$ <br> 1: Set $t = 0, \Lambda = \mathbf{0}_{m \times m}, H = \frac{1}{\alpha}\boldsymbol{I}_m$ and $V$ <br>     to any $m \times d$ matrix with orthonormal rows. <br> 2: Return $(\mathbf{0}_{m \times d}, H)$. |
| **SketchUpdate**$(\widehat{g})$ <br> 1: Insert $\widehat{g}$ into the last row of $S$. <br> 2: Compute eigendecomposition: $V^\top \Sigma V = S^\top S$ and set $S = (\Sigma - \Sigma_{m,m}\boldsymbol{I}_m)^{\frac{1}{2}}V$. <br> 3: Set $H = \mathrm{diag}\left\{\frac{1}{\alpha + \Sigma_{1,1} - \Sigma_{m,m}}, \cdots, \frac{1}{\alpha}\right\}$. <br> 4: Return $(S, H)$. | **SketchUpdate**$(\widehat{g})$ <br> 1: Update $t \leftarrow t+1$, $\Lambda$ and $V$ as Eqn. 4. <br> 2: Set $S = (t\Lambda)^{\frac{1}{2}}V$. <br> 3: Set $H = \mathrm{diag}\left\{\frac{1}{\alpha + t\Lambda_{1,1}}, \cdots, \frac{1}{\alpha + t\Lambda_{m,m}}\right\}$. <br> 4: Return $(S, H)$. |

**Random Projection (RP).** Random projections are classical methods for sketching [19, 1, 21]. Here we consider Gaussian Random Projection sketch: $S_t = S_{t-1} + \boldsymbol{r}_t\widehat{\boldsymbol{g}}_t^\top$, where each entry of $\boldsymbol{r}_t \in \mathbb{R}^m$ is an independent random Gaussian variable drawn from $\mathcal{N}(0, 1/\sqrt{m})$. One can verify that the update of $H_t^{-1}$ can be realized by two rank-one updates: $H_t^{-1} = H_{t-1}^{-1} + \boldsymbol{q}_t\boldsymbol{r}_t^\top + \boldsymbol{r}_t\boldsymbol{q}_t^\top$ where $\boldsymbol{q}_t = S_t\widehat{\boldsymbol{g}}_t - \frac{\|\widehat{\boldsymbol{g}}_t\|_2^2}{2}\boldsymbol{r}_t$. Using Woodbury formula, this results in $\mathcal{O}(md)$ update of $S$ and $H$ (see Algorithm 6 in Appendix E). We call this combination of SON with RP-sketch RP-SON. When $\alpha = 0$ this algorithm is invariant to linear transformations for each fixed realization of the randomness.

Using the existing guarantees for RP-sketch, in Appendix E we show a similar regret bound as Theorem 2 up to constants, provided $m = \tilde{\Omega}(r)$ where $r$ is the rank of $G_T$. Therefore RP-SON is near invariant, and gives substantial computational gains when $r \ll d$ with small regret overhead.

**Frequent Directions (FD).** When $G_T$ is near full-rank, however, RP-SON may not perform well. To address this, we consider Frequent Directions (FD) sketch [12, 23], a deterministic sketching method. FD maintains the invariant that the last row of $S_t$ is always $\mathbf{0}$. On each round, the vector $\widehat{\boldsymbol{g}}_t^\top$ is inserted into the last row of $S_{t-1}$, then the covariance of the resulting matrix is eigendecomposed into $V_t^\top \Sigma_t V_t$ and $S_t$ is set to $(\Sigma_t - \rho_t\boldsymbol{I}_m)^{\frac{1}{2}}V_t$ where $\rho_t$ is the smallest eigenvalue. Since the rows of $S_t$ are orthogonal to each other, $H_t$ is a diagonal matrix and can be maintained efficiently (see Algorithm 2). The sketch update works in $\mathcal{O}(md)$ time (see [12] and Appendix G.2) so the total running time is $\mathcal{O}(md)$ per round. We call this combination FD-SON and prove the following regret bound with notation $\Omega_k = \sum_{i=k+1}^d \lambda_i(G_T^\top G_T)$ for any $k = 0, \ldots, m-1$.

**Theorem 3.** *Under Assumptions 1 and 2, suppose that $\sigma_t \geq \sigma \geq 0$ for all $t$ and $\eta_t$ is non-increasing. FD-SON ensures that for any $\boldsymbol{w} \in \mathcal{K}$ and $k = 0, \ldots, m-1$, we have*

$$R_T(\boldsymbol{w}) \leq \frac{\alpha}{2}\|\boldsymbol{w}\|_2^2 + 2(CL)^2\sum_{t=1}^T \eta_t + \frac{m}{2(\sigma + \eta_T)}\ln\left(1 + \frac{\mathrm{TR}(S_T^\top S_T)}{m\alpha}\right) + \frac{m\Omega_k}{2(m-k)(\sigma+\eta_T)\alpha}.$$

Instead of the rank, the bound depends on the spectral decay $\Omega_k$, which essentially is the only extra term compared to the bound in Theorem 2. Similarly to previous discussion, if $\sigma_t \geq \sigma$, we get the bound $\frac{\alpha}{2}\|w\|_2^2 + \frac{m}{2\sigma}\ln\left(1 + \frac{\mathrm{TR}(S_T^\top S_T)}{m\alpha}\right) + \frac{m\Omega_k}{2(m-k)\sigma\alpha}$. With $\alpha$ tuned well, we pay logarithmic regret for the top $m$ eigenvectors, but a square root regret $\mathcal{O}(\sqrt{\Omega_k})$ for remaining directions not controlled by our sketch. This is expected for deterministic sketching which focuses on the dominant part of the spectrum. When $\alpha$ is not tuned we still get sublinear regret as long as $\Omega_k$ is sublinear.

**Oja's Algorithm.** Oja's algorithm [28, 29] is not usually considered as a sketching algorithm but seems very natural here. This algorithm uses online gradient descent to find eigenvectors and eigenvalues of data in a streaming fashion, with the to-sketch vector $\widehat{\boldsymbol{g}}_t$'s as the input. Specifically, let $V_t \in \mathbb{R}^{m \times d}$ denote the estimated eigenvectors and the diagonal matrix $\Lambda_t \in \mathbb{R}^{m \times m}$ contain the estimated eigenvalues at the end of round $t$. Oja's algorithm updates as:

$$\Lambda_t = (\boldsymbol{I}_m - \Gamma_t)\Lambda_{t-1} + \Gamma_t \, \mathrm{diag}\{V_{t-1}\widehat{\boldsymbol{g}}_t\}^2, \qquad V_t \xleftarrow{\mathrm{orth}} V_{t-1} + \Gamma_t V_{t-1}\widehat{\boldsymbol{g}}_t\widehat{\boldsymbol{g}}_t^\top \qquad (4)$$

where $\Gamma_t \in \mathbb{R}^{m \times m}$ is a diagonal matrix with (possibly different) learning rates of order $\Theta(1/t)$ on the diagonal, and the "$\xleftarrow{\text{orth}}$" operator represents an orthonormalizing step.[4] The sketch is then $S_t = (t\Lambda_t)^{\frac{1}{2}}V_t$. The rows of $S_t$ are orthogonal and thus $H_t$ is an efficiently maintainable diagonal matrix (see Algorithm 3). We call this combination Oja-SON.

The time complexity of Oja's algorithm is $\mathcal{O}(m^2 d)$ per round due to the orthonormalizing step. To improve the running time to $\mathcal{O}(md)$, one can only update the sketch every $m$ rounds (similar to the block power method [16, 22]). The regret guarantee of this algorithm is unclear since existing analysis for Oja's algorithm is only for the stochastic setting (see e.g. [2, 22]). However, Oja-SON provides good performance experimentally.

## 4 Sparse Implementation

In many applications, examples (and hence gradients) are sparse in the sense that $\|\boldsymbol{x}_t\|_0 \leq s$ for all $t$ and some small constant $s \ll d$. Most online first order methods enjoy a per-example running time depending on $s$ instead of $d$ in such settings. Achieving the same for second order methods is more difficult since $A_t^{-1}\boldsymbol{g}_t$ (or sketched versions) are typically dense even if $\boldsymbol{g}_t$ is sparse.

We show how to implement our algorithms in sparsity-dependent time, specifically, in $\mathcal{O}(m^2 + ms)$ for RP-SON and FD-SON and in $\mathcal{O}(m^3 + ms)$ for Oja-SON. We emphasize that since the sketch would still quickly become a dense matrix even if the examples are sparse, achieving purely sparsity-dependent time is highly non-trivial (especially for FD-SON and Oja-SON), and may be of independent interest. Due to space limit, below we only briefly mention how to do it for Oja-SON. Similar discussion for the other two sketches can be found in Appendix G. Note that mathematically these updates are equivalent to the non-sparse counterparts and regret guarantees are thus unchanged.

There are two ingredients to doing this for Oja-SON: (1) The eigenvectors $V_t$ are represented as $V_t = F_t Z_t$, where $Z_t \in \mathbb{R}^{m \times d}$ is a sparsely updatable direction (Step 3 in Algorithm 5) and $F_t \in \mathbb{R}^{m \times m}$ is a matrix such that $F_t Z_t$ is orthonormal. (2) The weights $\boldsymbol{w}_t$ are split as $\bar{\boldsymbol{w}}_t + Z_{t-1}^\top \boldsymbol{b}_t$, where $\boldsymbol{b}_t \in \mathbb{R}^m$ maintains the weights on the subspace captured by $V_{t-1}$ (same as $Z_{t-1}$), and $\bar{\boldsymbol{w}}_t$ captures the weights on the complementary subspace which are again updated sparsely.

We describe the sparse updates for $\bar{\boldsymbol{w}}_t$ and $\boldsymbol{b}_t$ below with the details for $F_t$ and $Z_t$ deferred to Appendix H. Since $S_t = (t\Lambda_t)^{\frac{1}{2}}V_t = (t\Lambda_t)^{\frac{1}{2}}F_t Z_t$ and $\boldsymbol{w}_t = \bar{\boldsymbol{w}}_t + Z_{t-1}^\top \boldsymbol{b}_t$, we know $\boldsymbol{u}_{t+1}$ is

$$\boldsymbol{w}_t - \left(\boldsymbol{I}_d - S_t^\top H_t S_t\right)\frac{\boldsymbol{g}_t}{\alpha} = \underbrace{\bar{\boldsymbol{w}}_t - \frac{\boldsymbol{g}_t}{\alpha} - (Z_t - Z_{t-1})^\top \boldsymbol{b}_t}_{\overset{\text{def}}{=} \bar{\boldsymbol{u}}_{t+1}} + Z_t^\top \underbrace{\left(\boldsymbol{b}_t + \tfrac{1}{\alpha}F_t^\top(t\Lambda_t H_t)F_t Z_t \boldsymbol{g}_t\right)}_{\overset{\text{def}}{=} \boldsymbol{b}'_{t+1}}. \quad (5)$$

Since $Z_t - Z_{t-1}$ is sparse by construction and the matrix operations defining $\boldsymbol{b}'_{t+1}$ scale with $m$, overall the update can be done in $\mathcal{O}(m^2 + ms)$. Using the update for $\boldsymbol{w}_{t+1}$ in terms of $\boldsymbol{u}_{t+1}$, $\boldsymbol{w}_{t+1}$ is equal to

$$\boldsymbol{u}_{t+1} - \gamma_t(\boldsymbol{I}_d - S_t^\top H_t S_t)\boldsymbol{x}_{t+1} = \underbrace{\bar{\boldsymbol{u}}_{t+1} - \gamma_t \boldsymbol{x}_{t+1}}_{\overset{\text{def}}{=} \bar{\boldsymbol{w}}_{t+1}} + Z_t^\top \underbrace{\left(\boldsymbol{b}'_{t+1} + \gamma_t F_t^\top(t\Lambda_t H_t)F_t Z_t \boldsymbol{x}_{t+1}\right)}_{\overset{\text{def}}{=} \boldsymbol{b}_{t+1}}. \quad (6)$$

Again, it is clear that all the computations scale with $s$ and not $d$, so both $\bar{\boldsymbol{w}}_{t+1}$ and $\boldsymbol{b}_{t+1}$ require only $O(m^2 + ms)$ time to maintain. Furthermore, the prediction $\boldsymbol{w}_t^\top \boldsymbol{x}_t = \bar{\boldsymbol{w}}_t^\top \boldsymbol{x}_t + \boldsymbol{b}_t^\top Z_{t-1}\boldsymbol{x}_t$ can also be computed in $\mathcal{O}(ms)$ time. The $\mathcal{O}(m^3)$ in the overall complexity comes from a Gram-Schmidt step in maintaining $F_t$ (details in Appendix H).

The pseudocode is presented in Algorithms 4 and 5 with some details deferred to Appendix H. This is the first sparse implementation of online eigenvector computation to the best of our knowledge.

## 5 Experiments

Preliminary experiments revealed that out of our three sketching options, Oja's sketch generally has better performance (see Appendix I). For more thorough evaluation, we implemented the sparse

**Algorithm 4** Sparse Sketched Online Newton with Oja's Algorithm
---
**Input:** Parameters $C$, $\alpha$ and $m$.

1: Initialize $\bar{\boldsymbol{u}} = \mathbf{0}_{d \times 1}$ and $\boldsymbol{b} = \mathbf{0}_{m \times 1}$.
2: $(\Lambda, F, Z, H) \leftarrow$ **SketchInit**$(\alpha, m)$    (Algorithm 5).
3: **for** $t = 1$ **to** $T$ **do**
4:     Receive example $\boldsymbol{x}_t$.
5:     **Projection step:** compute $\widehat{\boldsymbol{x}} = FZ\boldsymbol{x}_t$ and $\gamma = \frac{\tau_C(\bar{\boldsymbol{u}}^\top \boldsymbol{x}_t + \boldsymbol{b}^\top Z \boldsymbol{x}_t)}{\boldsymbol{x}_t^\top \boldsymbol{x}_t - (t-1)\widehat{\boldsymbol{x}}^\top \Lambda H \widehat{\boldsymbol{x}}}$.
         Obtain $\bar{\boldsymbol{w}} = \bar{\boldsymbol{u}} - \gamma \boldsymbol{x}_t$ and $\boldsymbol{b} \leftarrow \boldsymbol{b} + \gamma(t-1)F^\top \Lambda H \widehat{\boldsymbol{x}}$    (Equation 6).
6:     Predict label $y_t = \bar{\boldsymbol{w}}^\top \boldsymbol{x}_t + \boldsymbol{b}^\top Z \boldsymbol{x}_t$ and suffer loss $\ell_t(y_t)$.
7:     Compute gradient $\boldsymbol{g}_t = \ell_t'(y_t)\boldsymbol{x}_t$ and the *to-sketch vector* $\widehat{\boldsymbol{g}} = \sqrt{\sigma_t + \eta_t}\boldsymbol{g}_t$.
8:     $(\Lambda, F, Z, H, \boldsymbol{\delta}) \leftarrow$ **SketchUpdate**$(\widehat{\boldsymbol{g}})$    (Algorithm 5).
9:     **Update weight:** $\bar{\boldsymbol{u}} = \bar{\boldsymbol{w}} - \frac{1}{\alpha}\boldsymbol{g}_t - (\boldsymbol{\delta}^\top \boldsymbol{b})\widehat{\boldsymbol{g}}$ and $\boldsymbol{b} \leftarrow \boldsymbol{b} + \frac{1}{\alpha}tF^\top \Lambda H F Z \boldsymbol{g}_t$    (Equation 5).
10: **end for**
---

---
**Algorithm 5** Sparse Oja's Sketch
---
**Internal State:** $t$, $\Lambda$, $F$, $Z$, $H$ and $K$.

**SketchInit**$(\alpha, m)$
1: Set $t = 0, \Lambda = \mathbf{0}_{m \times m}, F = K = \alpha H = \boldsymbol{I}_m$ and $Z$ to any $m \times d$ matrix with orthonormal rows.
2: Return $(\Lambda, F, Z, H)$.

**SketchUpdate**$(\widehat{\boldsymbol{g}})$
1: Update $t \leftarrow t+1$. Pick a diagonal stepsize matrix $\Gamma_t$ to update $\Lambda \leftarrow (\boldsymbol{I} - \Gamma_t)\Lambda + \Gamma_t \, \text{diag}\{FZ\widehat{\boldsymbol{g}}\}^2$.
2: Set $\boldsymbol{\delta} = A^{-1}\Gamma_t FZ\widehat{\boldsymbol{g}}$ and update $K \leftarrow K + \boldsymbol{\delta}\widehat{\boldsymbol{g}}^\top Z^\top + Z\widehat{\boldsymbol{g}}\boldsymbol{\delta}^\top + (\widehat{\boldsymbol{g}}^\top \widehat{\boldsymbol{g}})\boldsymbol{\delta}\boldsymbol{\delta}^\top$.
3: Update $Z \leftarrow Z + \boldsymbol{\delta}\widehat{\boldsymbol{g}}^\top$.
4: $(L, Q) \leftarrow$ Decompose$(F, K)$ (Algorithm 13), so that $LQZ = FZ$ and $QZ$ is orthogonal. Set $F = Q$.
5: Set $H \leftarrow \text{diag}\left\{\frac{1}{\alpha + t\Lambda_{1,1}}, \cdots, \frac{1}{\alpha + t\Lambda_{m,m}}\right\}$.
6: Return $(\Lambda, F, Z, H, \boldsymbol{\delta})$.
---

version of Oja-SON in Vowpal Wabbit.[5] We compare it with ADAGRAD [6, 25] on both synthetic and real-world datasets. Each algorithm takes a stepsize parameter: $\frac{1}{\alpha}$ serves as a stepsize for Oja-SON and a scaling constant on the gradient matrix for ADAGRAD. We try both methods with the parameter set to $2^j$ for $j = -3, -2, \ldots, 6$ and report the best results. We keep the stepsize matrix in Oja-SON fixed as $\Gamma_t = \frac{1}{t}\boldsymbol{I}_m$ throughout. All methods make one online pass over data minimizing square loss.

## 5.1 Synthetic Datasets

To investigate Oja-SON's performance in the setting it is really designed for, we generated a range of synthetic ill-conditioned datasets as follows. We picked a random Gaussian matrix $Z \sim \mathbb{R}^{T \times d}$ ($T = 10{,}000$ and $d = 100$) and a random orthonormal basis $V \in \mathbb{R}^{d \times d}$. We chose a specific spectrum $\boldsymbol{\lambda} \in \mathbb{R}^d$ where the first $d - 10$ coordinates are 1 and the rest increase linearly to some fixed *condition number parameter* $\kappa$. We let $X = Z\text{diag}\{\boldsymbol{\lambda}\}^{\frac{1}{2}} V^\top$ be our example matrix, and created a binary classification problem with labels $y = \text{sign}(\boldsymbol{\theta}^\top \boldsymbol{x})$, where $\boldsymbol{\theta} \in \mathbb{R}^d$ is a random vector. We generated 20 such datasets with the same $Z$, $V$ and labels $y$ but different values of $\kappa \in \{10, 20, \ldots, 200\}$. Note that if the algorithm is truly invariant, it would have the same behavior on these 20 datasets.

Fig. 1 (in Section 1) shows the final progressive error (i.e. fraction of misclassified examples after one pass over data) for ADAGRAD and Oja-SON (with sketch size $m = 0, 5, 10$) as the condition number increases. As expected, the plot confirms the performance of first order methods such as ADAGRAD degrades when the data is ill-conditioned. The plot also shows that as the sketch size increases, Oja-SON becomes more accurate: when $m = 0$ (no sketch at all), Oja-SON is vanilla gradient descent and is worse than ADAGRAD as expected; when $m = 5$, the accuracy greatly improves; and finally when $m = 10$, the accuracy of Oja-SON is substantially better and hardly worsens with $\kappa$.

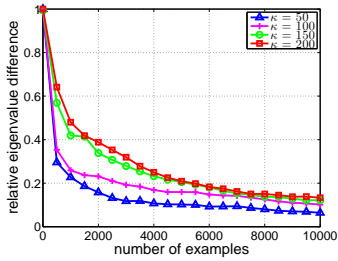

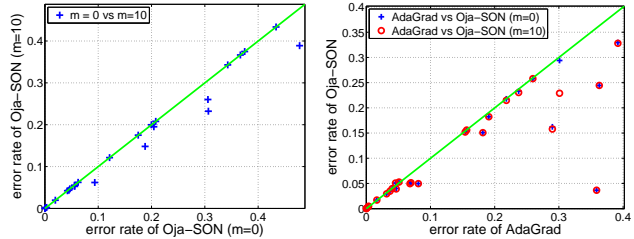

Figure 2: Oja's algorithm's eigenvalue recovery error.

Figure 3: (a) Comparison of two sketch sizes on real data, and (b) Comparison against ADAGRAD on real data.

To further explain the effectiveness of Oja's algorithm in identifying top eigenvalues and eigenvectors, the plot in Fig. 2 shows the largest relative difference between the true and estimated top 10 eigenvalues as Oja's algorithm sees more data. This gap drops quickly after seeing just 500 examples.

## 5.2 Real-world Datasets

Next we evaluated Oja-SON on 23 benchmark datasets from the UCI and LIBSVM repository (see Appendix I for description of these datasets). Note that some datasets are very high dimensional but very sparse (e.g. for *20news*, $d \approx 102,000$ and $s \approx 94$), and consequently methods with running time quadratic (such as ONS) or even linear in dimension rather than sparsity are prohibitive.

In Fig. 3(a), we show the effect of using sketched second order information, by comparing sketch size $m = 0$ and $m = 10$ for Oja-SON (concrete error rates in Appendix I). We observe significant improvements in 5 datasets (*acoustic, census, heart, ionosphere, letter*), demonstrating the advantage of using second order information. However, we found that Oja-SON was outperformed by ADA-GRAD on most datasets, mostly because the diagonal adaptation of ADAGRAD greatly reduces the condition number on these datasets. Moreover, one disadvantage of SON is that for the directions not in the sketch, it is essentially doing vanilla gradient descent. We expect better results using diagonal adaptation as in ADAGRAD in off-sketch directions.

To incorporate this high level idea, we performed a simple modification to Oja-SON: upon seeing example $\boldsymbol{x}_t$, we feed $D_t^{-\frac{1}{2}}\boldsymbol{x}_t$ to our algorithm instead of $\boldsymbol{x}_t$, where $D_t \in \mathbb{R}^{d \times d}$ is the diagonal part of the matrix $\sum_{\tau=1}^{t-1} \boldsymbol{g}_\tau \boldsymbol{g}_\tau^\top$.[6] The intuition is that this diagonal rescaling first homogenizes the scales of all dimensions. Any remaining ill-conditioning is further addressed by the sketching to some degree, while the complementary subspace is no worse-off than with ADAGRAD. We believe this flexibility in picking the right vectors to sketch is an attractive aspect of our sketching-based approach.

With this modification, Oja-SON outperforms ADAGRAD on most of the datasets even for $m = 0$, as shown in Fig. 3(b) (concrete error rates in Appendix I). The improvement on ADAGRAD at $m = 0$ is surprising but not impossible as the updates are not identical–our update is scale invariant like Ross et al. [33]. However, the diagonal adaptation already greatly reduces the condition number on all datasets except *splice* (see Fig. 4 in Appendix I for detailed results on this dataset), so little improvement is seen for sketch size $m = 10$ over $m = 0$. For several datasets, we verified the accuracy of Oja's method in computing the top-few eigenvalues (Appendix I), so the lack of difference between sketch sizes is due to the lack of second order information after the diagonal correction.

The average running time of our algorithm when $m = 10$ is about 11 times slower than ADAGRAD, matching expectations. Overall, SON can significantly outperform baselines on ill-conditioned data, while maintaining a practical computational complexity.

**Acknowledgements** This work was done when Haipeng Luo and Nicolò Cesa-Bianchi were at Microsoft Research, New York.

## Footnotes

[1]Recent work by [13] also studies sparse updates for a more complicated variant of Frequent Directions which is randomized and incurs extra approximation error.

[2]Stochastic setting assumes that the examples are drawn i.i.d. from a distribution.

[3]In the standard setting where $\boldsymbol{w}_t$ and $\boldsymbol{x}_t$ are restricted such that $\|\boldsymbol{w}_t\| \le D$ and $\|\boldsymbol{x}_t\| \le X$, the minimax regret is $\mathcal{O}(DXL\sqrt{T})$. This is clearly a special case of our setting with $C = DX$.

[4]For simplicity, we assume that $V_{t-1} + \Gamma_t V_{t-1}\widehat{\boldsymbol{g}}_t\widehat{\boldsymbol{g}}_t^\top$ is always of full rank so that the orthonormalizing step does not reduce the dimension of $V_t$.

[5]An open source machine learning toolkit available at http://hunch.net/~vw

[6]$D_1$ is defined as $0.1 \times \boldsymbol{I}_d$ to avoid division by zero.

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
