[Supplementary Material · appendix.pdf]

# Supplementary material for "Efficient Second Order Online Learning by Sketching"

## A Proof of Theorem 1

*Proof.* Assuming $T$ is a multiple of $d$ without loss of generality, we pick $\boldsymbol{x}_t$ from the basis vectors $\{\boldsymbol{e}_1, \ldots, \boldsymbol{e}_d\}$ so that each $\boldsymbol{e}_i$ appears $T/d$ times (in an arbitrary order). Note that now $\mathcal{K}$ is just a hypercube:
$$\mathcal{K} = \left\{\boldsymbol{w} \,:\, |\boldsymbol{w}^\top \boldsymbol{x}_t| \leq C, \ \ \forall t\right\} = \left\{\boldsymbol{w} \,:\, \|\boldsymbol{w}\|_\infty \leq C\right\}.$$

Let $\xi_1, \ldots, \xi_T$ be independent Rademacher random variables such that $\Pr(\xi_t = +1) = \Pr(\xi_t = -1) = \frac{1}{2}$. For a scalar $\theta$, we define loss function[7] $\ell_t(\theta) = (\xi_t L)\theta$, so that Assumptions 1 and 2 are clearly satisfied with $\sigma_t = 0$. We show that, for any online algorithm,
$$\mathbb{E}[R_T] = \mathbb{E}\left[\sum_{t=1}^{T} \ell_t(\boldsymbol{w}_t^\top \boldsymbol{x}_t) - \inf_{\boldsymbol{w} \in \mathcal{K}} \sum_{t=1}^{T} \ell_t(\boldsymbol{w}^\top \boldsymbol{x}_t)\right] \geq CL\sqrt{\frac{dT}{2}}$$

which implies the statement of the theorem.

First of all, note that $\mathbb{E}\left[\ell_t(\boldsymbol{w}_t^\top \boldsymbol{x}_t) \,\middle|\, \xi_1, \ldots, \xi_{t-1}\right] = 0$ for any $\boldsymbol{w}_t$. Hence we have

$$\mathbb{E}\left[\sum_{t=1}^{T} \ell_t(\boldsymbol{w}_t^\top \boldsymbol{x}_t) - \inf_{\boldsymbol{w} \in \mathcal{K}} \sum_{t=1}^{T} \ell_t(\boldsymbol{w}^\top \boldsymbol{x}_t)\right] = \mathbb{E}\left[\sup_{\boldsymbol{w} \in \mathcal{K}} \sum_{t=1}^{T} -\ell_t(\boldsymbol{w}^\top \boldsymbol{x}_t)\right] = L\,\mathbb{E}\left[\sup_{\boldsymbol{w} \in \mathcal{K}} \boldsymbol{w}^\top \sum_{t=1}^{T} \xi_t \boldsymbol{x}_t\right],$$

which, by the construction of $\boldsymbol{x}_t$, is

$$CL\,\mathbb{E}\left[\left\|\sum_{t=1}^{T} \xi_t \boldsymbol{x}_t\right\|_1\right] = CLd\,\mathbb{E}\left[\left|\sum_{t=1}^{T/d} \xi_t\right|\right] \geq CLd\sqrt{\frac{T}{2d}} = CL\sqrt{\frac{dT}{2}},$$

where the final bound is due to the Khintchine inequality (see e.g. Lemma 8.2 in [4]). This concludes the proof. $\qquad\square$

## B Projection

We prove a more general version of Lemma 1 which does not require invertibility of the matrix $A$ here.

**Lemma 2.** *For any $\boldsymbol{x} \neq \boldsymbol{0}, \boldsymbol{u} \in \mathbb{R}^{d \times 1}$ and positive semidefinite matrix $A \in \mathbb{R}^{d \times d}$, we have*

$$\boldsymbol{w}^* = \operatorname*{argmin}_{\boldsymbol{w} : |\boldsymbol{w}^\top \boldsymbol{x}| \leq C} \|\boldsymbol{w} - \boldsymbol{u}\|_A = \begin{cases} \boldsymbol{u} - \dfrac{\tau_C(\boldsymbol{u}^\top \boldsymbol{x})}{\boldsymbol{x}^\top A^\dagger \boldsymbol{x}} A^\dagger \boldsymbol{x} & \text{if } \boldsymbol{x} \in \operatorname{range}(A) \\[2mm] \boldsymbol{u} - \dfrac{\tau_C(\boldsymbol{u}^\top \boldsymbol{x})}{\boldsymbol{x}^\top (\boldsymbol{I} - A^\dagger A)\boldsymbol{x}} (\boldsymbol{I} - A^\dagger A)\boldsymbol{x} & \text{if } \boldsymbol{x} \notin \operatorname{range}(A) \end{cases}$$

*where $\tau_C(y) = \operatorname{SGN}(y) \max\{|y| - C, 0\}$ and $A^\dagger$ is the Moore-Penrose pseudoinverse of $A$. (Note that when $A$ is rank deficient, this is one of the many possible solutions.)*

*Proof.* First consider the case when $\boldsymbol{x} \in \operatorname{range}(A)$. If $|\boldsymbol{u}^\top \boldsymbol{x}| \leq C$, then it is trivial that $\boldsymbol{w}^* = \boldsymbol{u}$. We thus assume $\boldsymbol{u}^\top \boldsymbol{x} \geq C$ below (the last case $\boldsymbol{u}^\top \boldsymbol{x} \leq -C$ is similar). The Lagrangian of the problem is
$$L(\boldsymbol{w}, \lambda_1, \lambda_2) = \frac{1}{2}(\boldsymbol{w} - \boldsymbol{u})^\top A(\boldsymbol{w} - \boldsymbol{u}) + \lambda_1(\boldsymbol{w}^\top \boldsymbol{x} - C) + \lambda_2(\boldsymbol{w}^\top \boldsymbol{x} + C)$$

where $\lambda_1 \geq 0$ and $\lambda_2 \leq 0$ are Lagrangian multipliers. Since $\boldsymbol{w}^\top \boldsymbol{x}$ cannot be $C$ and $-C$ at the same time, The complementary slackness condition implies that either $\lambda_1 = 0$ or $\lambda_2 = 0$. Suppose the latter case is true, then setting the derivative with respect to $\boldsymbol{w}$ to 0, we get $\boldsymbol{w}^* = \boldsymbol{u} - \lambda_1 A^\dagger \boldsymbol{x} + (\boldsymbol{I} - A^\dagger A)\boldsymbol{z}$

where $\boldsymbol{z} \in R^{d \times 1}$ can be arbitrary. However, since $A(\boldsymbol{I} - A^\dagger A) = 0$, this part does not affect the objective value at all and we can simply pick $z = 0$ so that $w^*$ has a consistent form regardless of whether $A$ is full rank or not. Now plugging $\boldsymbol{w}^*$ back, we have

$$L(\boldsymbol{w}^*, \lambda_1, 0) = -\frac{\lambda_1^2}{2} \boldsymbol{x}^\top A^\dagger \boldsymbol{x} + \lambda_1 (\boldsymbol{u}^\top \boldsymbol{x} - C)$$

which is maximized when $\lambda_1 = \frac{\boldsymbol{u}^\top \boldsymbol{x} - C}{\boldsymbol{x}^\top A^\dagger \boldsymbol{x}} \geq 0$. Plugging this optimal $\lambda_1$ into $\boldsymbol{w}^*$ gives the stated solution. On the other hand, if $\lambda_1 = 0$ instead, we can proceed similarly and verify that it gives a smaller dual value (0 in fact), proving the previous solution is indeed optimal.

We now move on to the case when $\boldsymbol{x} \notin \mathrm{range}(A)$. First of all the stated solution is well defined since $\boldsymbol{x}^\top (\boldsymbol{I} - A^\dagger A) \boldsymbol{x}$ is nonzero in this case. Moreover, direct calculation shows that $\boldsymbol{w}^*$ is in the valid space: $|\boldsymbol{w}^{*\top} \boldsymbol{x}| = |\boldsymbol{u}^\top \boldsymbol{x} - \tau_C(\boldsymbol{u}^\top \boldsymbol{x})| \leq C$, and also it gives the minimal possible distance value $\|\boldsymbol{w}^* - \boldsymbol{u}\|_A = 0$, proving the lemma. $\qquad\square$

## C   Proof of Theorem 2

We first prove a general regret bound that holds for any choice of $A_t$ in update 1:

$$\boldsymbol{u}_{t+1} = \boldsymbol{w}_t - A_t^{-1} \boldsymbol{g}_t$$
$$\boldsymbol{w}_{t+1} = \underset{\boldsymbol{w} \in \mathcal{K}_{t+1}}{\mathrm{argmin}} \|\boldsymbol{w} - \boldsymbol{u}_{t+1}\|_{A_t} .$$

This bound will also be useful in proving regret guarantees for the sketched versions.

**Proposition 1.** *For any sequence of positive definite matrices $A_t$ and sequence of losses satisfying Assumptions 1 and 2, the regret of updates* (1) *against any comparator $\boldsymbol{w} \in \mathcal{K}$ satisfies*

$$2R_T(\boldsymbol{w}) \leq \|\boldsymbol{w}\|_{A_0}^2 + \underbrace{\sum_{t=1}^{T} \boldsymbol{g}_t^T A_t^{-1} \boldsymbol{g}_t}_{\text{``Gradient Bound'' } R_G} + \underbrace{\sum_{t=1}^{T} (\boldsymbol{w}_t - \boldsymbol{w})^\top (A_t - A_{t-1} - \sigma_t \boldsymbol{g}_t \boldsymbol{g}_t^\top)(\boldsymbol{w}_t - \boldsymbol{w})}_{\text{``Diameter Bound'' } R_D}$$

*Proof.* Since $\boldsymbol{w}_{t+1}$ is the projection of $\boldsymbol{u}_{t+1}$ onto $\mathcal{K}_{t+1}$, by the property of projections (see for example [17, Lemma 8]), the algorithm ensures

$$\|\boldsymbol{w}_{t+1} - \boldsymbol{w}\|_{A_t}^2 \leq \|\boldsymbol{u}_{t+1} - \boldsymbol{w}\|_{A_t}^2 = \|\boldsymbol{w}_t - \boldsymbol{w}\|_{A_t}^2 + \boldsymbol{g}_t^\top A_t^{-1} \boldsymbol{g}_t - 2\boldsymbol{g}_t^\top (\boldsymbol{w}_t - \boldsymbol{w})$$

for all $\boldsymbol{w} \in \mathcal{K} \subseteq \mathcal{K}_{t+1}$. By the curvature property in Assumption 2, we then have that

$$
\begin{aligned}
2R_T(\boldsymbol{w}) &\leq \sum_{t=1}^{T} 2\boldsymbol{g}_t^\top (\boldsymbol{w}_t - \boldsymbol{w}) - \sigma_t \big( \boldsymbol{g}_t^\top (\boldsymbol{w}_t - \boldsymbol{w}) \big)^2 \\
&\leq \sum_{t=1}^{T} \boldsymbol{g}_t^\top A_t^{-1} \boldsymbol{g}_t + \|\boldsymbol{w}_t - \boldsymbol{w}\|_{A_t}^2 - \|\boldsymbol{w}_{t+1} - \boldsymbol{w}\|_{A_t}^2 - \sigma_t \big( \boldsymbol{g}_t^\top (\boldsymbol{w}_t - \boldsymbol{w}) \big)^2 \\
&\leq \|\boldsymbol{w}\|_{A_0}^2 + \sum_{t=1}^{T} \boldsymbol{g}_t^\top A_t^{-1} \boldsymbol{g}_t + (\boldsymbol{w}_t - \boldsymbol{w})^\top (A_t - A_{t-1} - \sigma_t \boldsymbol{g}_t \boldsymbol{g}_t^\top)(\boldsymbol{w}_t - \boldsymbol{w}),
\end{aligned}
$$

which completes the proof. $\qquad\square$

*Proof of Theorem 2.* We apply Proposition 1 with the choice: $A_0 = \alpha \boldsymbol{I}_d$ and $A_t = A_{t-1} + (\sigma_t + \eta_t)\boldsymbol{g}_t \boldsymbol{g}_t^T$, which gives $\|\boldsymbol{w}\|_{A_0}^2 = \alpha \|\boldsymbol{w}\|_2^2$ and

$$R_D = \sum_{t=1}^{T} \eta_t (\boldsymbol{w}_t - \boldsymbol{w})^\top \boldsymbol{g}_t \boldsymbol{g}_t^\top (\boldsymbol{w}_t - \boldsymbol{w}) \leq 4(CL)^2 \sum_{t=1}^{T} \eta_t ,$$

where the last equality uses the Lipschitz property in Assumption 1 and the boundedness of $\boldsymbol{w}_t^\top \boldsymbol{x}_t$ and $\boldsymbol{w}^\top \boldsymbol{x}_t$.

For the term $R_G$, define $\widehat{A}_t = \frac{\alpha}{\sigma+\eta_T}\boldsymbol{I}_d + \sum_{s=1}^t \boldsymbol{g}_s\boldsymbol{g}_s^\top$. Since $\sigma_t \geq \sigma$ and $\eta_t$ is non-increasing, we have $\widehat{A}_t \preceq \frac{1}{\sigma+\eta_T}A_t$, and therefore:

$$
\begin{aligned}
R_G &\leq \frac{1}{\sigma+\eta_T}\sum_{t=1}^T \boldsymbol{g}_t^\top \widehat{A}_t^{-1}\boldsymbol{g}_t = \frac{1}{\sigma+\eta_T}\sum_{t=1}^T \left\langle \widehat{A}_t - \widehat{A}_{t-1},\ \widehat{A}_t^{-1}\right\rangle \\
&\leq \frac{1}{\sigma+\eta_T}\sum_{t=1}^T \ln\frac{|\widehat{A}_t|}{|\widehat{A}_{t-1}|} = \frac{1}{\sigma+\eta_T}\ln\frac{|\widehat{A}_T|}{|\widehat{A}_0|} \\
&= \frac{1}{\sigma+\eta_T}\sum_{i=1}^d \ln\left(1 + \frac{(\sigma+\eta_T)\lambda_i\left(\sum_{t=1}^T \boldsymbol{g}_t\boldsymbol{g}_t^\top\right)}{\alpha}\right) \\
&\leq \frac{d}{\sigma+\eta_T}\ln\left(1 + \frac{(\sigma+\eta_T)\sum_{i=1}^d \lambda_i\left(\sum_{t=1}^T \boldsymbol{g}_t\boldsymbol{g}_t^\top\right)}{d\alpha}\right) \\
&= \frac{d}{\sigma+\eta_T}\ln\left(1 + \frac{(\sigma+\eta_T)\sum_{t=1}^T \|\boldsymbol{g}_t\|_2^2}{d\alpha}\right)
\end{aligned}
$$

where the second inequality is by the concavity of the function $\ln|X|$ (see [18, Lemma 12] for an alternative proof), and the last one is by Jensen's inequality. This concludes the proof. $\qquad\square$

## D  A Truly Invariant Algorithm

In this section we discuss how to make our adaptive online Newton algorithm truly invariant to invertible linear transformations. To achieve this, we set $\alpha = 0$ and replace $A_t^{-1}$ with the Moore-Penrose pseudoinverse $A_t^\dagger$:[8]

$$
\begin{aligned}
\boldsymbol{u}_{t+1} &= \boldsymbol{w}_t - A_t^\dagger \boldsymbol{g}_t, \\
\boldsymbol{w}_{t+1} &= \operatorname*{argmin}_{\boldsymbol{w}\in\mathcal{K}_{t+1}} \|\boldsymbol{w} - \boldsymbol{u}_{t+1}\|_{A_t}.
\end{aligned}
\tag{7}
$$

When written in this form, it is not immediately clear that the algorithm has the invariant property. However, one can rewrite the algorithm in a mirror descent form:

$$
\begin{aligned}
\boldsymbol{w}_{t+1} &= \operatorname*{argmin}_{\boldsymbol{w}\in\mathcal{K}_{t+1}} \left\|\boldsymbol{w} - \boldsymbol{w}_t + A_t^\dagger \boldsymbol{g}_t\right\|_{A_t}^2 \\
&= \operatorname*{argmin}_{\boldsymbol{w}\in\mathcal{K}_{t+1}} \|\boldsymbol{w} - \boldsymbol{w}_t\|_{A_t}^2 + 2(\boldsymbol{w} - \boldsymbol{w}_t)^\top A_t A_t^\dagger \boldsymbol{g}_t \\
&= \operatorname*{argmin}_{\boldsymbol{w}\in\mathcal{K}_{t+1}} \|\boldsymbol{w} - \boldsymbol{w}_t\|_{A_t}^2 + 2\boldsymbol{w}^\top \boldsymbol{g}_t
\end{aligned}
$$

where we use the fact that $\boldsymbol{g}_t$ is in the range of $A_t$ in the last step. Now suppose all the data $\boldsymbol{x}_t$ are transformed to $M\boldsymbol{x}_t$ for some unknown and invertible matrix $M$, then one can verify that all the weights will be transformed to $M^{-T}\boldsymbol{w}_t$ accordingly, ensuring the prediction to remain the same.

Moreover, the regret bound of this algorithm can be bounded as below. First notice that even when $A_t$ is rank deficient, the projection step still ensures the following: $\|\boldsymbol{w}_{t+1} - \boldsymbol{w}\|_{A_t}^2 \leq \|\boldsymbol{u}_{t+1} - \boldsymbol{w}\|_{A_t}^2$, which is proven in [18, Lemma 8]. Therefore, the entire proof of Theorem 2 still holds after replacing $A_t^{-1}$ with $A_t^\dagger$, giving the regret bound:

$$
\frac{1}{2}\sum_{t=1}^T \boldsymbol{g}_t^\top A_t^\dagger \boldsymbol{g}_t + 2(CL)^2\eta_t.
\tag{8}
$$

The key now is to bound the term $\sum_{t=1}^T \boldsymbol{g}_t^\top \widehat{A}_t^\dagger \boldsymbol{g}_t$ where we define $\widehat{A}_t = \sum_{s=1}^t \boldsymbol{g}_s\boldsymbol{g}_s^\top$. In order to do this, we proceed similarly to the proof of [5, Theorem 4.2] to show that this term is of order $\mathcal{O}(d^2\ln T)$ in the worst case.

**Theorem 4.** *Let $\lambda^*$ be the minimum among the smallest nonzero eigenvalues of $\widehat{A}_t$ ($t = 1, \dots, T$) and $r$ be the rank of $\widehat{A}_T$. We have*

$$\sum_{t=1}^{T} \boldsymbol{g}_t^\top \widehat{A}_t^\dagger \boldsymbol{g}_t \leq r + \frac{(1+r)r}{2} \ln \left( 1 + \frac{2 \sum_{t=1}^{T} \|\boldsymbol{g}_t\|_2^2}{(1+r)r\lambda^*} \right) .$$

*Proof.* First by Cesa-Bianchi et al. [5, Lemma D.1], we have

$$\boldsymbol{g}_t^\top \widehat{A}_t^\dagger \boldsymbol{g}_t = \begin{cases} 1 & \text{if } \boldsymbol{g}_t \notin \text{range}(\widehat{A}_{t-1}) \\ 1 - \frac{\det_+(\widehat{A}_{t-1})}{\det_+(\widehat{A}_t)} < 1 & \text{if } \boldsymbol{g}_t \in \text{range}(\widehat{A}_{t-1}) \end{cases}$$

where $\det_+(M)$ denotes the product of the nonzero eigenvalues of matrix $M$. We thus separate the steps $t$ such that $\boldsymbol{g}_t \in \text{range}(\widehat{A}_{t-1})$ from those where $\boldsymbol{g}_t \notin \text{range}(\widehat{A}_{t-1})$. For each $k = 1, \dots, r$ let $T_k$ be the first time step $t$ in which the rank of $A_t$ is $k$ (so that $T_1 = 1$). Also let $T_{r+1} = T + 1$ for convenience. With this notation, we have

$$\sum_{t=1}^{T} \boldsymbol{g}_t^\top \widehat{A}_t^\dagger \boldsymbol{g}_t = \sum_{k=1}^{r} \left( \boldsymbol{g}_{T_k}^\top \widehat{A}_{T_k}^\dagger \boldsymbol{g}_{T_k} + \sum_{t=T_k+1}^{T_{k+1}-1} \boldsymbol{g}_t^\top \widehat{A}_t^\dagger \boldsymbol{g}_t \right)$$

$$= \sum_{k=1}^{r} \left( 1 + \sum_{t=T_k+1}^{T_{k+1}-1} \left( 1 - \frac{\det_+(\widehat{A}_{t-1})}{\det_+(\widehat{A}_t)} \right) \right)$$

$$= r + \sum_{k=1}^{r} \sum_{t=T_k+1}^{T_{k+1}-1} \left( 1 - \frac{\det_+(\widehat{A}_{t-1})}{\det_+(\widehat{A}_t)} \right)$$

$$\leq r + \sum_{k=1}^{r} \sum_{t=T_k+1}^{T_{k+1}-1} \ln \frac{\det_+(\widehat{A}_t)}{\det_+(\widehat{A}_{t-1})}$$

$$= r + \sum_{k=1}^{r} \ln \frac{\det_+(\widehat{A}_{T_{k+1}-1})}{\det_+(\widehat{A}_{T_k})} .$$

Fix any $k$ and let $\lambda_{k,1}, \dots, \lambda_{k,k}$ be the nonzero eigenvalues of $\widehat{A}_{T_k}$ and $\lambda_{k,1} + \mu_{k,1}, \dots, \lambda_{k,k} + \mu_{k,k}$ be the nonzero eigenvalues of $\widehat{A}_{T_{k+1}-1}$. Then

$$\ln \frac{\det_+(\widehat{A}_{T_{k+1}-1})}{\det_+(\widehat{A}_{T_k})} = \ln \prod_{i=1}^{k} \frac{\lambda_{k,i} + \mu_{k,i}}{\lambda_{k,i}} = \sum_{i=1}^{k} \ln \left( 1 + \frac{\mu_{k,i}}{\lambda_{k,i}} \right) .$$

Hence, we arrive at

$$\sum_{t=1}^{T} \boldsymbol{g}_t^\top \widehat{A}_t^+ \boldsymbol{g}_t \leq r + \sum_{k=1}^{r} \sum_{i=1}^{k} \ln \left( 1 + \frac{\mu_{k,i}}{\lambda_{k,i}} \right) .$$

To further bound the latter quantity, we use $\lambda^* \leq \lambda_{k,i}$ and Jensen's inequality :

$$\sum_{k=1}^{r} \sum_{i=1}^{k} \ln \left( 1 + \frac{\mu_{k,i}}{\lambda_{k,i}} \right) \leq \sum_{k=1}^{r} \sum_{i=1}^{k} \ln \left( 1 + \frac{\mu_{k,i}}{\lambda^*} \right)$$

$$\leq \frac{(1+r)r}{2} \ln \left( 1 + \frac{2 \sum_{k=1}^{r} \sum_{i=1}^{k} \mu_{k,i}}{(1+r)r\lambda^*} \right) .$$

Finally noticing that

$$\sum_{i=1}^{k} \mu_{k,i} = \text{TR}(\widehat{A}_{T_{k+1}-1}) - \text{TR}(\widehat{A}_{T_k}) = \sum_{t=T_k+1}^{T_{k+1}-1} \text{TR}(\boldsymbol{g}_t \boldsymbol{g}_t^\top) = \sum_{t=T_k+1}^{T_{k+1}-1} \|\boldsymbol{g}_t\|_2^2$$

completes the proof. □

---

**Algorithm 6** Random Projection Sketch for RP-SON

---
**Internal State:** $S$ and $H$.

**SketchInit**$(\alpha, m)$
  1: Set $S = \mathbf{0}_{m \times d}$ and $H = \frac{1}{\alpha} \boldsymbol{I}_m$.
  2: Return $(S, H)$.

**SketchUpdate**$(\widehat{g})$
  1: Draw $\boldsymbol{r} \sim \mathcal{N}(0, \frac{1}{\sqrt{m}})$ and update $S \leftarrow S + \boldsymbol{r}\widehat{\boldsymbol{g}}^\top$.
  2: Compute $\boldsymbol{q} = S\widehat{\boldsymbol{g}} - \frac{\widehat{\boldsymbol{g}}^\top \widehat{\boldsymbol{g}}}{2}\boldsymbol{r}$, update $H \leftarrow H - \frac{H\boldsymbol{q}\boldsymbol{r}^\top H}{1+\boldsymbol{r}^\top H\boldsymbol{q}}$ and $H \leftarrow H - \frac{H\boldsymbol{r}\boldsymbol{q}^\top H}{1+\boldsymbol{q}^\top H\boldsymbol{r}}$.
  3: Return $(S, H)$.

---

Taken together, Eq. (8) and Theorem 4 lead to the following regret bounds (recall the definitions of $\lambda^*$ and $r$ from Theorem 4).

**Corollary 1.** *If $\sigma_t = 0$ for all $t$ and $\eta_t$ is set to be $\frac{1}{CL}\sqrt{\frac{d}{t}}$, then the regret of the algorithm defined by Eq. (7) is at most*

$$\frac{CL}{2}\sqrt{\frac{T}{d}}\left(r + \frac{(1+r)r}{2}\ln\left(1 + \frac{2\sum_{t=1}^{T}\|\boldsymbol{g}_t\|_2^2}{(1+r)r\lambda^*}\right)\right) + 4CL\sqrt{Td}.$$

*On the other hand, if $\sigma_t \geq \sigma > 0$ for all $t$ and $\eta_t$ is set to be $0$, then the regret is at most*

$$\frac{1}{2\sigma}\left(r + \frac{(1+r)r}{2}\ln\left(1 + \frac{2\sum_{t=1}^{T}\|\boldsymbol{g}_t\|_2^2}{(1+r)r\lambda^*}\right)\right).$$

## E  Regret Bound for RP-SON

The pseudocode of the RP sketch is presented in Algorithm 6. Recall the notation $R_G$ and $R_D$ in Proposition 1 and let $r$ be the rank of $G_T$, we prove the following regret bound:

**Theorem 5.** *Under Assumptions 1 and 2, if the sketch size $m = \Omega\big((r + \ln(T/\delta))\epsilon^{-2}\big)$, then RP-SON ensures*
*(1) $\mathbb{E}[R_D] \leq 4(CL)^2 \sum_{t=1}^{T} \eta_t$, and*
*(2) $R_G \leq \frac{1}{1-\epsilon} \sum_{t=1}^{T} \boldsymbol{g}_t^\top (\alpha \boldsymbol{I}_d + G_t^\top G_t)^{-1} \boldsymbol{g}_t$ with probability at least $1 - \delta$.*

*Proof.* We apply the property of the random projection method (see for example [36, Theorem 2.3]): as long as $m = \Omega\big((r + \ln(T/\delta))\epsilon^{-2}\big)$, with probability at least $1 - \delta$,

$$(1 - \epsilon)G_t^\top G_t \preceq S_t^\top S_t \preceq (1 + \epsilon)G_t^\top G_t \qquad \text{for all } t = 1, \ldots, T$$

which implies $A_t^{-1} \preceq \frac{1}{1-\epsilon}(\alpha \boldsymbol{I}_d + G_t^\top G_t)^{-1}$ and thus $R_G \leq \frac{1}{1-\epsilon} \sum_{t=1}^{T} \boldsymbol{g}_t^\top (\alpha \boldsymbol{I}_d + G_t^\top G_t)^{-1} \boldsymbol{g}_t$. For $R_D$, first fix all the randomness before drawing $\boldsymbol{r}_t$ and let $\mathbb{E}_t$ be the corresponding conditional expectation, then we have

$$\mathbb{E}_t[A_t - A_{t-1}] = \mathbb{E}_t\left[S_{t-1}^\top \boldsymbol{r}_t \widehat{\boldsymbol{g}}_t^\top + \widehat{\boldsymbol{g}}_t \boldsymbol{r}_t^\top S_{t-1} + \|\boldsymbol{r}_t\|_2^2 \widehat{\boldsymbol{g}}_t \widehat{\boldsymbol{g}}_t^\top\right] = (\sigma_t + \eta_t)\boldsymbol{g}_t \boldsymbol{g}_t^\top.$$

Since $\boldsymbol{w}_t, \boldsymbol{w}$ and $\boldsymbol{g}_t$ are fixed, we continue with

$$\mathbb{E}_t\left[(\boldsymbol{w}_t - \boldsymbol{w})^\top (A_t - A_{t-1} - \sigma_t \boldsymbol{g}_t \boldsymbol{g}_t^\top)(\boldsymbol{w}_t - \boldsymbol{w})\right] = \eta_t(\boldsymbol{w}_t - \boldsymbol{w})^\top \boldsymbol{g}_t \boldsymbol{g}_t^\top (\boldsymbol{w}_t - \boldsymbol{w}) \leq 4(CL)^2 \eta_t.$$

Therefore, taking the overall expectation gives $\mathbb{E}[R_D] \leq 4(CL)^2 \sum_{t=1}^{T} \eta_t$. $\qquad\square$

This theorem implies that the bound on $R_D$ is the same as the one without using sketch, and the term $R_G$ is only constant larger.

# F  Proof of Theorem 3

*Proof.* We again first apply Proposition 1 (recall the notation $R_G$ and $R_D$ stated in the proposition). By the construction of the sketch, we have

$$A_t - A_{t-1} = S_t^\top S_t - S_{t-1}^\top S_{t-1} = \widehat{g}_t \widehat{g}_t^\top - \rho_t V_t^\top V_t \preceq \widehat{g}_t \widehat{g}_t^\top \; .$$

It follows immediately that $R_D$ is again at most $4(CL)^2 \sum_{t=1}^T \eta_t$. For the term $R_G$, we will apply the following guarantee of Frequent Directions (see the proof of Theorem 1.1 of [12]): $\sum_{t=1}^T \rho_t \leq \frac{\Omega_k}{m-k}$. Specifically, since $\mathrm{TR}(V_t A_t^{-1} V_t^\top) \leq \frac{1}{\alpha}\mathrm{TR}(V_t V_t^\top) = \frac{m}{\alpha}$ we have

$$
\begin{aligned}
R_G &= \sum_{t=1}^T \frac{1}{\sigma_t + \eta_t} \left\langle A_t^{-1}, A_t - A_{t-1} + \rho_t V_t^\top V_t \right\rangle \\
&\leq \frac{1}{\sigma + \eta_T} \sum_{t=1}^T \left( \left\langle A_t^{-1}, A_t - A_{t-1} + \rho_t V_t^\top V_t \right\rangle \right) \\
&= \frac{1}{\sigma + \eta_T} \sum_{t=1}^T \left( \left\langle A_t^{-1}, A_t - A_{t-1} \right\rangle + \rho_t \mathrm{TR}(V_t A_t^{-1} V_t^\top) \right) \\
&\leq \frac{1}{(\sigma + \eta_T)} \sum_{t=1}^T \left\langle A_t^{-1}, A_t - A_{t-1} \right\rangle + \frac{m\Omega_k}{(m-k)(\sigma + \eta_T)\alpha} \; .
\end{aligned}
$$

Finally for the term $\sum_{t=1}^T \left\langle A_t^{-1}, A_t - A_{t-1} \right\rangle$, we proceed similarly to the proof of Theorem 2:

$$
\begin{aligned}
\sum_{t=1}^T \left\langle A_t^{-1}, A_t - A_{t-1} \right\rangle &\leq \sum_{t=1}^T \ln \frac{|A_t|}{|A_{t-1}|} = \ln \frac{|A_T|}{|A_0|} = \sum_{i=1}^d \ln \left(1 + \frac{\lambda_i(S_T^\top S_T)}{\alpha}\right) \\
&= \sum_{i=1}^m \ln \left(1 + \frac{\lambda_i(S_T^\top S_T)}{\alpha}\right) \leq m \ln \left(1 + \frac{\mathrm{TR}(S_T^\top S_T)}{m\alpha}\right)
\end{aligned}
$$

where the first inequality is by the concavity of the function $\ln |X|$, the second one is by Jensen's inequality, and the last equality is by the fact that $S_T^\top S_T$ is of rank $m$ and thus $\lambda_i(S_T^\top S_T) = 0$ for any $i > m$. This concludes the proof. □

# G  Sparse updates for RP and FD sketches

## G.1  Random Projection

We recall the updates of RP sketch. Since $\widehat{g}_t$ is sparse, $S_t = S_{t-1} + r\widehat{g}_t^\top$ is easily updated in $\mathcal{O}(ms)$ time. $H_t$ can also be updated in $\mathcal{O}(m^2 + ms)$ time clearly. However, since the sketch $S_t$ is getting denser and denser, direct update of the weight vector is a dense operation too. The solution is to represent and store $w_t$ in the form of $\bar{w}_t + S_{t-1}^\top b_t$ for some $\bar{w}_t \in \mathbb{R}^d$ and $b_t \in \mathbb{R}^m$. Note that now computing the prediction $w_t^\top x_t$ needs $\mathcal{O}(ms)$ time. Rewriting the update rules we have

$$
\begin{aligned}
u_{t+1} &= w_t - \frac{1}{\alpha} g_t + \frac{1}{\alpha} S_t^\top H_t S_t g_t = \bar{w}_t + S_{t-1}^\top b_t - \frac{1}{\alpha} g_t + \frac{1}{\alpha} S_t^\top H_t S_t g_t \\
&= \underbrace{\bar{w}_t - \widehat{g}_t r_t^\top b_t - \frac{1}{\alpha} g_t}_{\stackrel{\text{def}}{=} \bar{u}_{t+1}} + S_t^\top \underbrace{(b_t + \frac{1}{\alpha} H_t S_t g_t)}_{\stackrel{\text{def}}{=} b'_{t+1}} \; .
\end{aligned}
$$

Since $g_t$ and $\widehat{g}_t$ are sparse, computing $\bar{u}_{t+1}$ and $b'_{t+1}$ needs $\mathcal{O}(m^2 + ms)$ time. Finally, for the projection step, $c_t$ can clearly be computed in $\mathcal{O}(m^2 + ms)$ time, and the update rule of $\bar{w}_{t+1}$ and

**Algorithm 7** Sparse Sketched Online Newton with Random Projection

---

**Input:** Parameters $C$, $\alpha$ and $m$.

1: Initialize $\bar{\boldsymbol{u}} = \boldsymbol{0}_{d\times 1}$, $\boldsymbol{b} = \boldsymbol{0}_{m\times 1}$ and $(S, H) \leftarrow$ **SketchInit**$(\alpha, m)$ (Algorithm 6).
2: **for** $t = 1$ **to** $T$ **do**
3: $\quad$ Receive example $\boldsymbol{x}_t$.
4: $\quad$ Projection step: compute $\widehat{\boldsymbol{x}} = S\boldsymbol{x}_t$, $\gamma = \frac{\tau_C(\bar{\boldsymbol{u}}^\top \boldsymbol{x}_t + \boldsymbol{b}^\top \widehat{\boldsymbol{x}})}{\boldsymbol{x}_t^\top \boldsymbol{x}_t - \widehat{\boldsymbol{x}}^\top H\widehat{\boldsymbol{x}}}$, $\bar{\boldsymbol{w}} = \bar{\boldsymbol{u}} - \gamma\boldsymbol{x}_t$ and $\boldsymbol{b} \leftarrow \boldsymbol{b} + cH\widehat{\boldsymbol{x}}$.
5: $\quad$ Predict label $y_t = \bar{\boldsymbol{w}}^\top \boldsymbol{x}_t + \boldsymbol{b}^\top \widehat{\boldsymbol{x}}$ and suffer loss $\ell_t(y_t)$.
6: $\quad$ Compute gradient $\boldsymbol{g}_t = \ell_t'(y_t)\boldsymbol{x}_t$ and the to-sketch vector $\widehat{\boldsymbol{g}} = \sqrt{\sigma_t + \eta_t}\boldsymbol{g}_t$.
7: $\quad$ $(S, H) \leftarrow$ **SketchUpdate**$(\widehat{\boldsymbol{g}})$ (Algorithm 6).
8: $\quad$ Update $\bar{\boldsymbol{u}} = \bar{\boldsymbol{w}} - (\boldsymbol{r}^\top \boldsymbol{b})\widehat{\boldsymbol{g}} - \frac{1}{\alpha}\boldsymbol{g}_t$ and $\boldsymbol{b} \leftarrow \boldsymbol{b} + \frac{1}{\alpha}HS\boldsymbol{g}_t$.
9: **end for**

---

$\boldsymbol{b}_{t+1}$ is thus derived as follows:

$$
\begin{aligned}
\boldsymbol{w}_{t+1} &= \boldsymbol{u}_{t+1} - \gamma_t(\boldsymbol{x}_{t+1} - S_t^\top H_t S_t \boldsymbol{x}_{t+1}) \\
&= \bar{\boldsymbol{u}}_{t+1} + S_t^\top \boldsymbol{b}_{t+1}' - \gamma_t(\boldsymbol{x}_{t+1} - S_t^\top H_t S_t \boldsymbol{x}_{t+1}) \\
&= \underbrace{\bar{\boldsymbol{u}}_{t+1} - \gamma_t\boldsymbol{x}_{t+1}}_{\overset{\text{def}}{=}\bar{\boldsymbol{w}}_{t+1}} + S_t^\top \underbrace{(\boldsymbol{b}_{t+1}' + \gamma_t H_t S_t \boldsymbol{x}_{t+1})}_{\overset{\text{def}}{=}\boldsymbol{b}_{t+1}}
\end{aligned}
$$

which again takes $\mathcal{O}(m^2 + ms)$ time. Taken together, the total time complexity per round is $\mathcal{O}(m^2 + ms)$. The pseudocode for this version of the algorithm is presented in Algorithm 7.

## G.2 Frequent Directions

The sparse version of our algorithm with the Frequent Directions option is much more involved. We begin by taking a detour and introducing a fast and epoch-based variant of the Frequent Directions algorithm proposed in [12]. The idea is the following: instead of doing an eigendecomposition immediately after inserting a new $\widehat{\boldsymbol{g}}$ every round, we double the size of the sketch (to $2m$), keep up to $m$ recent $\widehat{\boldsymbol{g}}$'s, do the decomposition only at the end of every $m$ rounds and finally keep the top $m$ eigenvectors with shrunk eigenvalues. The advantage of this variant is that it can be implemented straightforwardly in $\mathcal{O}(md)$ time on average without doing a complicated rank-one SVD update, while still ensuring the exact same guarantee with the only price of doubling the sketch size.

Algorithm 8 shows the details of this variant and how we maintain $H$. The sketch $S$ is always represented by two parts: the top part $(DV)$ comes from the last eigendecomposition, and the bottom part $(G)$ collects the recent to-sketch vector $\widehat{\boldsymbol{g}}$'s. Note that within each epoch, the update of $H^{-1}$ is a rank-two update and thus $H$ can be updated similarly to the case of random projection (Lines 3 and 4 of Algorithm 8).

Although we can use any available algorithm that runs in $\mathcal{O}(m^2 d)$ time to do the eigendecomposition (Line 7 in Algorithm 8), we explicitly write down the procedure of reducing this problem to eigendecomposing a small square matrix in Algorithm 9, which will be important for deriving the sparse version of the algorithm. Lemma 3 proves that Algorithm 9 works correctly for finding the top $m$ eigenvector and eigenvalues.

**Lemma 3.** *The outputs of Algorithm 9 are such that the $i$-th row of $V'$ and the $i$-th entry of the diagonal of $\Sigma$ are the $i$-th eigenvector and eigenvalue of $S^\top S$ respectively.*

*Proof.* Let $W^\top \in \mathbb{R}^{d\times(d-m-r)}$ be an orthonormal basis of the null space of $\begin{pmatrix} V \\ Q \end{pmatrix}$. By Line 2, we know that $GW^\top = \boldsymbol{0}$ and $E = (V^\top \ Q^\top \ W^\top)$ forms an orthonormal basis of $\mathbb{R}^d$. Therefore, we

---

**Algorithm 8** Frequent Direction Sketch (epoch version)

---

**Internal State:** $\tau, D, V, G$ and $H$.

**SketchInit($\alpha, m$)**
1: Set $\tau = 1, D = \mathbf{0}_{m \times m}, G = \mathbf{0}_{m \times d}, H = \frac{1}{\alpha} \boldsymbol{I}_{2m}$ and let $V$ be any $m \times d$ matrix whose rows are orthonormal.
2: Return $(\mathbf{0}_{2m \times d}, H)$.

**SketchUpdate($\widehat{g}$)**
1: Insert $\widehat{g}$ into the $\tau$-th row of $G$.
2: **if** $\tau < m$ **then**
3:      Let $e$ be the $2m \times 1$ basis vector whose $(m+\tau)$-th entry is 1 and $q = S\widehat{g} - \frac{\widehat{g}^\top \widehat{g}}{2} e$.
4:      Update $H \leftarrow H - \frac{Hqe^\top H}{1 + e^\top Hq}$ and $H \leftarrow H - \frac{Heq^\top H}{1 + q^\top He}$.
5:      Update $\tau \leftarrow \tau + 1$.
6: **else**
7:      $(V, \Sigma) \leftarrow$ **ComputeEigenSystem** $\left( \begin{pmatrix} DV \\ G \end{pmatrix} \right)$ (Algorithm 9).
8:      Set $D$ to be a diagonal matrix with $D_{i,i} = \sqrt{\Sigma_{i,i} - \Sigma_{m,m}}, \ \forall i \in [m]$.
9:      Set $H \leftarrow \text{diag} \left\{ \frac{1}{\alpha + D_{1,1}^2}, \cdots, \frac{1}{\alpha + D_{m,m}^2}, \frac{1}{\alpha}, \ldots, \frac{1}{\alpha} \right\}$.
10:     Set $G = \mathbf{0}_{m \times d}$.
11:     Set $\tau = 1$.
12: **end if**
13: Return $\left( \begin{pmatrix} DV \\ G \end{pmatrix}, H \right)$.

---

---

**Algorithm 9** ComputeEigenSystem($S$)

---

**Input:** $S = \begin{pmatrix} DV \\ G \end{pmatrix}$.

**Output:** $V' \in \mathbb{R}^{m \times d}$ and diagonal matrix $\Sigma \in \mathbb{R}^{m \times m}$ such that the $i$-th row of $V'$ and the $i$-th entry of the diagonal of $\Sigma$ are the $i$-th eigenvector and eigenvalue of $S^\top S$ respectively.
1: Compute $M = GV^\top$.
2: Decompose $G - MV$ into the form $LQ$ where $L \in \mathbb{R}^{m \times r}$, $Q$ is a $r \times d$ matrix whose rows are orthonormal and $r$ is the rank of $G - MV$ (e.g. by a Gram-Schmidt process).
3: Compute the top $m$ eigenvectors ($U \in \mathbb{R}^{m \times (m+r)}$) and eigenvalues ($\Sigma \in \mathbb{R}^{m \times m}$) of the matrix
$$\begin{pmatrix} D^2 & \mathbf{0}_{m \times r} \\ \mathbf{0}_{r \times m} & \mathbf{0}_{r \times r} \end{pmatrix} + \begin{pmatrix} M^\top \\ L^\top \end{pmatrix} \begin{pmatrix} M & L \end{pmatrix}.$$
4: Return $(V', \Sigma)$ where $V' = U \begin{pmatrix} V \\ Q \end{pmatrix}$.

---

have

$$
\begin{aligned}
S^\top S &= V^\top D^2 V + G^\top G \\
&= E \begin{pmatrix} D^2 & \mathbf{0} & \mathbf{0} \\ \mathbf{0} & \mathbf{0} & \mathbf{0} \\ \mathbf{0} & \mathbf{0} & \mathbf{0} \end{pmatrix} E^\top + EE^\top G^\top GEE^\top \\
&= E \left( \begin{pmatrix} D^2 & \mathbf{0} & \mathbf{0} \\ \mathbf{0} & \mathbf{0} & \mathbf{0} \\ \mathbf{0} & \mathbf{0} & \mathbf{0} \end{pmatrix} + \begin{pmatrix} VG^\top \\ QG^\top \\ WG^\top \end{pmatrix} (GV^\top \ \ GQ^\top \ \ GW^\top) \right) E^\top \\
&= (V^\top \ Q^\top) \underbrace{\left( \begin{pmatrix} D^2 & \mathbf{0} \\ \mathbf{0} & \mathbf{0} \end{pmatrix} + \begin{pmatrix} M^\top \\ L^\top \end{pmatrix} (M \ \ L) \right)}_{=C} \begin{pmatrix} V \\ Q \end{pmatrix}
\end{aligned}
$$

where in the last step we use the fact $GQ^\top = (MV + LQ)Q^\top = L$. Now it is clear that the eigenvalue of $C$ will be the eigenvalue of $S^\top S$ and the eigenvector of $C$ will be the eigenvector of $S^\top S$ after left multiplied by matrix $(V^\top \; Q^\top)$, completing the proof. $\qquad\square$

We are now ready to present the sparse version of SON with Frequent Direction sketch (Algorithm 10). The key point is that we represent $V_t$ as $F_t Z_t$ for some $F_t \in \mathbb{R}^{m \times m}$ and $Z_t \in \mathbb{R}^{m \times d}$, and the weight vector $\boldsymbol{w}_t$ as $\bar{\boldsymbol{w}}_t + Z_{t-1}^\top \boldsymbol{b}_t$ and ensure that the update of $Z_t$ and $\bar{\boldsymbol{w}}_t$ will always be sparse. To see this, denote the sketch $S_t$ by $\begin{pmatrix} D_t F_t Z_t \\ G_t \end{pmatrix}$ and let $H_{t,1}$ and $H_{t,2}$ be the top and bottom half of $H_t$. Now the update rule of $\boldsymbol{u}_{t+1}$ can be rewritten as

$$\boldsymbol{u}_{t+1} = \boldsymbol{w}_t - \left(\boldsymbol{I}_d - S_t^\top H_t S_t\right)\frac{\boldsymbol{g}_t}{\alpha}$$

$$= \bar{\boldsymbol{w}}_t + Z_{t-1}^\top \boldsymbol{b}_t - \frac{1}{\alpha}\boldsymbol{g}_t + \frac{1}{\alpha}(Z_t^\top F_t^\top D_t, G_t^\top)\begin{pmatrix} H_{t,1}S_t\boldsymbol{g}_t \\ H_{t,2}S_t\boldsymbol{g}_t \end{pmatrix}$$

$$= \underbrace{\bar{\boldsymbol{w}}_t + \frac{1}{\alpha}(G_t^\top H_{t,2}S_t\boldsymbol{g}_t - \boldsymbol{g}_t) - (Z_t - Z_{t-1})^\top \boldsymbol{b}_t}_{\bar{\boldsymbol{u}}_{t+1}} + Z_t^\top \underbrace{(\boldsymbol{b}_t + \frac{1}{\alpha}F_t^\top D_t H_{t,1}S_t\boldsymbol{g}_t)}_{\boldsymbol{b}_{t+1}'}$$

We will show that $Z_t - Z_{t-1} = \Delta_t G_t$ for some $\Delta_t \in \mathbb{R}^{m \times m}$ shortly, and thus the above update is efficient due to the fact that the rows of $G_t$ are collections of previous sparse vectors $\hat{\boldsymbol{g}}$.

Similarly, the update of $\boldsymbol{w}_{t+1}$ can be written as

$$\boldsymbol{w}_{t+1} = \boldsymbol{u}_{t+1} - \gamma_t(\boldsymbol{x}_{t+1} - S_t^\top H_t S_t \boldsymbol{x}_{t+1})$$

$$= \bar{\boldsymbol{u}}_{t+1} + Z_t^\top \boldsymbol{b}_{t+1}' - \gamma_t \boldsymbol{x}_{t+1} + \gamma_t(Z_t^\top F_t^\top D_t, G_t^\top)\begin{pmatrix} H_{t,1}S_t\boldsymbol{x}_{t+1} \\ H_{t,2}S_t\boldsymbol{x}_{t+1} \end{pmatrix}$$

$$= \underbrace{\bar{\boldsymbol{u}}_{t+1} + \gamma_t(G_t^\top H_{t,2}S_t\boldsymbol{x}_{t+1} - \boldsymbol{x}_{t+1})}_{\bar{\boldsymbol{w}}_{t+1}} + Z_t^\top \underbrace{(\boldsymbol{b}_{t+1}' + \gamma_t F_t^\top D_t H_{t,1}S_t\boldsymbol{x}_{t+1})}_{\boldsymbol{b}_{t+1}}.$$

It is clear that $\gamma_t$ can be computed efficiently, and thus the update of $\boldsymbol{w}_{t+1}$ is also efficient. These updates correspond to Line 6 and 10 of Algorithm 10.

It remains to perform the sketch update efficiently. Algorithm 11 is the sparse version of Algorithm 8. The challenging part is to compute eigenvectors and eigenvalues efficiently. Fortunately, in light of Algorithm 9, using the new representation $V = FZ$ one can directly translate the process to Algorithm 12 and find that the eigenvectors can be expressed in the form $N_1 Z + N_2 G$. To see this, first note that Line 1 of both algorithms compute the same matrix $M = GV^\top = GZ^\top F^\top$. Then Line 2 decomposes the matrix

$$G - MV = G - MFZ = (\; -MF \quad \boldsymbol{I}_m\;)\begin{pmatrix} Z \\ G \end{pmatrix} \overset{\text{def}}{=} PR$$

using Gram-Schmidt into the form $LQR$ such that the rows of $QR$ are orthonormal (that is, $QR$ corresponds to $Q$ in Algorithm 9). While directly applying Gram-Schmidt to $PR$ would take $\mathcal{O}(m^2 d)$ time, this step can in fact be efficiently implemented by performing Gram-Schmidt to $P$ (instead of $PR$) in a Banach space where inner product is defined as $\langle \boldsymbol{a}, \boldsymbol{b}\rangle = \boldsymbol{a}^\top K \boldsymbol{b}$ with

$$K = RR^\top = \begin{pmatrix} ZZ^\top & ZG^\top \\ GZ^\top & GG^\top \end{pmatrix}$$

being the Gram matrix of $R$. Since we can efficiently maintain the Gram matrix of $Z$ (see Line 10 of Algorithm 11) and $GZ^\top$ and $GG^\top$ can be computed sparsely, this decomposing step can be done efficiently too. This modified Gram-Schmidt algorithm is presented in Algorithm 13 (which will also be used in sparse Oja's sketch), where Line 4 is the key difference compared to standard Gram-Schmidt (see Lemma 4 below for a formal proof of correctness).

Line 3 of Algorithms 9 and 12 are exactly the same. Finally the eigenvectors $U\begin{pmatrix} V \\ Q \end{pmatrix}$ in Algorithm 9 now becomes (with $U_1, U_2, Q_1, Q_2, N_1, N_2$ defined in Line 4 of Algorithm 12)

$$U\begin{pmatrix} FZ \\ QR \end{pmatrix} = (U_1, U_2)\begin{pmatrix} FZ \\ QR \end{pmatrix} = U_1 FZ + U_2(Q_1, Q_2)\begin{pmatrix} Z \\ G \end{pmatrix}$$

$$= (U_1 FZ + U_2 Q_1)Z + U_2 Q_2 G = N_1 Z + N_2 G.$$

---

**Algorithm 10** Sparse Sketched Online Newton with Frequent Directions

---

**Input:** Parameters $C$, $\alpha$ and $m$.

1: Initialize $\bar{\boldsymbol{u}} = \mathbf{0}_{d\times 1}$, $\boldsymbol{b} = \mathbf{0}_{m\times 1}$ and $(D, F, Z, G, H) \leftarrow$ **SketchInit**$(\alpha, m)$ (Algorithm 11).

2: Let $S$ denote the matrix $\begin{pmatrix} DFZ \\ G \end{pmatrix}$ throughout the algorithm (without actually computing it).

3: Let $H_1$ and $H_2$ denote the upper and lower half of $H$, i.e. $H = \begin{pmatrix} H_1 \\ H_2 \end{pmatrix}$.

4: **for** $t = 1$ **to** $T$ **do**

5:    Receive example $\boldsymbol{x}_t$.

6:    Projection step: compute $\widehat{\boldsymbol{x}} = S\boldsymbol{x}_t$ and $\gamma = \frac{\tau_C(\bar{\boldsymbol{u}}^\top \boldsymbol{x}_t + \boldsymbol{b}^\top Z\boldsymbol{x}_t)}{\boldsymbol{x}_t^\top \boldsymbol{x}_t - \widehat{\boldsymbol{x}}^\top H\widehat{\boldsymbol{x}}}$.
      Obtain $\bar{\boldsymbol{w}} = \bar{\boldsymbol{u}} + \gamma(G^\top H_2\widehat{\boldsymbol{x}} - \boldsymbol{x}_t)$ and $\boldsymbol{b} \leftarrow \boldsymbol{b} + \gamma F^\top DH_1\widehat{\boldsymbol{x}}$.

7:    Predict label $y_t = \bar{\boldsymbol{w}}^\top \boldsymbol{x}_t + \boldsymbol{b}^\top Z\boldsymbol{x}_t$ and suffer loss $\ell_t(y_t)$.

8:    Compute gradient $\boldsymbol{g}_t = \ell_t'(y_t)\boldsymbol{x}_t$ and the to-sketch vector $\widehat{\boldsymbol{g}} = \sqrt{\sigma_t + \eta_{lt}}\boldsymbol{g}_t$.

9:    $(D, F, Z, G, H, \Delta) \leftarrow$ **SketchUpdate**$(\widehat{\boldsymbol{g}})$ (Algorithm 11).

10:    Update $\bar{\boldsymbol{u}} = \bar{\boldsymbol{w}} + \frac{1}{\alpha}(G^\top H_2 S\boldsymbol{g} - \boldsymbol{g}) - G^\top \Delta^\top \boldsymbol{b}$ and $\boldsymbol{b} \leftarrow \boldsymbol{b} + \frac{1}{\alpha}F^\top DH_1 S\boldsymbol{g}$.

11: **end for**

---

---

**Algorithm 11** Sparse Frequent Direction Sketch

---

**Internal State:** $\tau, D, F, Z, G, H$ and $K$.

**SketchInit**$(\alpha, m)$

1: Set $\tau = 1$, $D = \mathbf{0}_{m\times m}$, $F = K = \boldsymbol{I}_m$, $H = \frac{1}{\alpha}\boldsymbol{I}_{2m}$, $G = \mathbf{0}_{m\times d}$, and let $Z$ be any $m \times d$ matrix whose rows are orthonormal.

2: Return $(D, F, Z, G, H)$.

**SketchUpdate**$(\widehat{\boldsymbol{g}})$

1: Insert $\widehat{\boldsymbol{g}}$ into the $\tau$-th row of $G$.

2: **if** $\tau < m$ **then**

3:    Let $\boldsymbol{e}$ be the $2m \times 1$ basic vector whose $(m + \tau)$-th entry is 1 and compute $\boldsymbol{q} = S\widehat{\boldsymbol{g}} - \frac{\widehat{\boldsymbol{g}}^\top \widehat{\boldsymbol{g}}}{2}\boldsymbol{e}$.

4:    Update $H \leftarrow H - \frac{H\boldsymbol{q}\boldsymbol{e}^\top H}{1+\boldsymbol{e}^\top H\boldsymbol{q}}$ and $H \leftarrow H - \frac{H\boldsymbol{e}\boldsymbol{q}^\top H}{1+\boldsymbol{q}^\top H\boldsymbol{e}}$.

5:    Set $\Delta = \mathbf{0}_{m\times m}$.

6:    Set $\tau \leftarrow \tau + 1$.

7: **else**

8:    $(N_1, N_2, \Sigma) \leftarrow$ **ComputeSparseEigenSystem**$\left(\begin{pmatrix} DFZ \\ G \end{pmatrix}, K\right)$ (Algorithm 12).

9:    Compute $\Delta = N_1^{-1}N_2$.

10:    Update Gram matrix $K \leftarrow K + \Delta GZ^\top + ZG^\top \Delta^\top + \Delta GG^\top \Delta^\top$.

11:    Update $F = N_1$, $Z \leftarrow Z + \Delta G$, and let $D$ be such that $D_{i,i} = \sqrt{\Sigma_{i,i} - \Sigma_{m,m}}$, $\forall i \in [m]$.

12:    Set $H \leftarrow \text{diag}\left\{\frac{1}{\alpha+D_{1,1}^2}, \cdots, \frac{1}{\alpha+D_{m,m}^2}, \frac{1}{\alpha}, \ldots, \frac{1}{\alpha}\right\}$.

13:    Set $G = \mathbf{0}_{m\times d}$.

14:    Set $\tau = 1$.

15: **end if**

16: Return $(D, F, Z, G, H, \Delta)$.

---

Therefore, having the eigenvectors in the form $N_1Z + N_2G$, we can simply update $F$ as $N_1$ and $Z$ as $Z + N_1^{-1}N_2G$ so that the invariant $V = FZ$ still holds (see Line 11 of Algorithm 11). The update of $Z$ is sparse since $G$ is sparse.

We finally summarize the results of this section in the following theorem.

**Theorem 6.** *The average running time of Algorithm 10 is $\mathcal{O}(m^2 + ms)$ per round, and the regret bound is exactly the same as the one stated in Theorem 3.*

**Lemma 4.** *The output of Algorithm 13 ensures that $LQR = PR$ and the rows of $QR$ are orthonormal.*

---

**Algorithm 12** ComputeSparseEigenSystem$(S, K)$

---

**Input:** $S = \begin{pmatrix} DFZ \\ G \end{pmatrix}$ and Gram matrix $K = ZZ^\top$.

**Output:** $N_1, N_2 \in \mathbb{R}^{m \times m}$ and diagonal matrix $\Sigma \in \mathbb{R}^{m \times m}$ such that the $i$-th row of $N_1 Z + N_2 G$ and the $i$-th entry of the diagonal of $\Sigma$ are the $i$-th eigenvector and eigenvalue of the matrix $S^\top S$.

1: Compute $M = GZ^\top F^\top$.

2: $(L, Q) \leftarrow$ Decompose $\left( \begin{pmatrix} -MF & \boldsymbol{I}_m \end{pmatrix}, \begin{pmatrix} K & ZG^\top \\ GZ^\top & GG^\top \end{pmatrix} \right)$ (Algorithm 13).

3: Let $r$ be the number of columns of $L$. Compute the top $m$ eigenvectors ($U \in \mathbb{R}^{m \times (m+r)}$) and eigenvalues ($\Sigma \in \mathbb{R}^{m \times m}$) of the matrix $\begin{pmatrix} D^2 & \boldsymbol{0}_{m \times r} \\ \boldsymbol{0}_{r \times m} & \boldsymbol{0}_{r \times r} \end{pmatrix} + \begin{pmatrix} M^\top \\ L^\top \end{pmatrix} \begin{pmatrix} M & L \end{pmatrix}$.

4: Set $N_1 = U_1 F + U_2 Q_1$ and $N_2 = U_2 Q_2$ where $U_1$ and $U_2$ are the first $m$ and last $r$ columns of $U$ respectively, and $Q_1$ and $Q_2$ are the left and right half of $Q$ respectively.

5: Return $(N_1, N_2, \Sigma)$.

---

*Proof.* It suffices to prove that Algorithm 13 is exactly the same as using the standard Gram-Schmidt to decompose the matrix $PR$ into $L$ and an orthonormal matrix which can be written as $QR$. First note that when $K = \boldsymbol{I}_n$, Algorithm 13 is simply the standard Gram-Schmidt algorithm applied to $P$. We will thus go through Line 1-10 of Algorithm 13 with $P$ replaced by $PR$ and $K$ by $\boldsymbol{I}_n$ and show that it leads to the exact same calculations as running Algorithm 13 directly. For clarity, we add "˜" to symbols to distinguish the two cases (so $\tilde{P} = PR$ and $\tilde{K} = \boldsymbol{I}_n$). We will inductively prove the invariance $\tilde{Q} = QR$ and $\tilde{L} = L$. The base case $\tilde{Q} = QR = \boldsymbol{0}$ and $\tilde{L} = L = \boldsymbol{0}$ is trivial. Now assume it holds for iteration $i - 1$ and consider iteration $i$. We have

$$\tilde{\boldsymbol{\alpha}} = \tilde{Q}\tilde{K}\tilde{\boldsymbol{p}} = QRR^\top \boldsymbol{p} = QK\boldsymbol{p} = \boldsymbol{\alpha},$$

$$\tilde{\boldsymbol{\beta}} = \tilde{\boldsymbol{p}} - \tilde{Q}^\top \tilde{\boldsymbol{\alpha}} = R^\top \boldsymbol{p} - (QR)^\top \boldsymbol{\alpha} = R^\top (\boldsymbol{p} - Q^\top \boldsymbol{\alpha}) = R^\top \boldsymbol{\beta},$$

$$\tilde{c} = \sqrt{\tilde{\boldsymbol{\beta}}^\top \tilde{K} \tilde{\boldsymbol{\beta}}} = \sqrt{(R^\top \boldsymbol{\beta})^\top (R^\top \boldsymbol{\beta})} = \sqrt{\boldsymbol{\beta}^\top K \boldsymbol{\beta}} = c,$$

which clearly implies that after execution of Line 5-9, we again have $\tilde{Q} = QR$ and $\tilde{L} = L$, finishing the induction. □

## H   Details for sparse Oja's algorithm

We finally provide the missing details for the sparse version of the Oja's algorithm. Since we already discussed the updates for $\bar{\boldsymbol{w}}_t$ and $\boldsymbol{b}_t$ in Section 4, we just need to describe how the updates for $F_t$ and $Z_t$ work. Recall that the dense Oja's updates can be written in terms of $F$ and $Z$ as

$$\begin{aligned} \Lambda_t &= (\boldsymbol{I}_m - \Gamma_t)\Lambda_{t-1} + \Gamma_t \operatorname{diag}\{F_{t-1}Z_{t-1}\widehat{\boldsymbol{g}}_t\}^2 \\ F_t Z_t &\xleftarrow{\text{orth}} F_{t-1}Z_{t-1} + \Gamma_t F_{t-1}Z_{t-1}\widehat{\boldsymbol{g}}_t\widehat{\boldsymbol{g}}_t^\top = F_{t-1}(Z_{t-1} + F_{t-1}^{-1}\Gamma_t F_{t-1}Z_{t-1}\widehat{\boldsymbol{g}}_t\widehat{\boldsymbol{g}}_t^\top). \end{aligned} \tag{9}$$

Here, the update for the eigenvalues is straightforward. For the update of eigenvectors, first we let $Z_t = Z_{t-1} + \boldsymbol{\delta}_t \widehat{\boldsymbol{g}}_t^\top$ where $\boldsymbol{\delta}_t = F_{t-1}^{-1}\Gamma_t F_{t-1}Z_{t-1}\widehat{\boldsymbol{g}}_t$ (note that under the assumption of Footnote 4, $F_t$ is always invertible). Now it is clear that $Z_t - Z_{t-1}$ is a sparse rank-one matrix and the update of $\bar{\boldsymbol{u}}_{t+1}$ is efficient. Finally it remains to update $F_t$ so that $F_t Z_t$ is the same as orthonormalizing $F_{t-1}Z_t$, which can in fact be achieved by applying the Gram-Schmidt algorithm to $F_{t-1}$ in a Banach space where inner product is defined as $\langle \boldsymbol{a}, \boldsymbol{b} \rangle = \boldsymbol{a}^\top K_t \boldsymbol{b}$ where $K_t$ is the Gram matrix $Z_t Z_t^\top$ (see Algorithm 13). Since we can maintain $K_t$ efficiently based on the update of $Z_t$:

$$K_t = K_{t-1} + \boldsymbol{\delta}_t \widehat{\boldsymbol{g}}_t^\top Z_{t-1}^\top + Z_{t-1}\widehat{\boldsymbol{g}}_t \boldsymbol{\delta}_t^\top + (\widehat{\boldsymbol{g}}_t^\top \widehat{\boldsymbol{g}}_t)\boldsymbol{\delta}_t \boldsymbol{\delta}_t^\top,$$

the update of $F_t$ can therefore be implemented in $\mathcal{O}(m^3)$ time.

---
**Algorithm 13** Decompose(P, K)

---
**Input:** $P \in \mathbb{R}^{m \times n}$, $K \in \mathbb{R}^{m \times m}$ such that $K$ is the Gram matrix $K = RR^\top$ for some matrix $R \in \mathbb{R}^{n \times d}$ where $n \geq m, d \geq m$,
**Output:** $L \in \mathbb{R}^{m \times r}$ and $Q \in \mathbb{R}^{r \times n}$ such that $LQR = PR$ where $r$ is the rank of $PR$ and the rows of $QR$ are orthonormal.
 1: Initialize $L = \mathbf{0}_{m \times m}$ and $Q = \mathbf{0}_{m \times n}$.
 2: **for** $i = 1$ **to** $m$ **do**
 3:     Let $\boldsymbol{p}^\top$ be the $i$-th row of $P$.
 4:     Compute $\boldsymbol{\alpha} = QK\boldsymbol{p}, \boldsymbol{\beta} = \boldsymbol{p} - Q^\top \boldsymbol{\alpha}$ and $c = \sqrt{\boldsymbol{\beta}^\top K \boldsymbol{\beta}}$.
 5:     **if** $c \neq 0$ **then**
 6:         Insert $\frac{1}{c}\boldsymbol{\beta}^\top$ to the $i$-th row of $Q$.
 7:     **end if**
 8:     Set the $i$-th entry of $\boldsymbol{\alpha}$ to be $c$ and insert $\boldsymbol{\alpha}$ to the $i$-th row of $L$.
 9: **end for**
10: Delete the all-zero columns of $L$ and all-zero rows of $Q$.
11: Return $(L, Q)$.

---

Figure 4: Error rates for Oja-SON with different sketch sizes on splice dataset

# I   Experiment Details

This section reports some detailed experimental results omitted from Section 5.2. Table 1 includes the description of benchmark datasets; Table 2 reports error rates on relatively small datasets to show that Oja-SON generally has better performance; Table 3 reports concrete error rates for the experiments described in Section 5.2; finally Table 4 shows that Oja's algorithm estimates the eigenvalues accurately.

As mentioned in Section 5.2, we see substantial improvement for the *splice* dataset when using Oja's sketch even after the diagonal adaptation. We verify that the condition number for this dataset before and after the diagonal adaptation are very close (682 and 668 respectively), explaining why a large improvement is seen using Oja's sketch. Fig. 4 shows the decrease of error rates as Oja-SON with different sketch sizes sees more examples. One can see that even with $m = 1$ Oja-SON already performs very well. This also matches our expectation since there is a huge gap between the top and second eigenvalues of this dataset ($50.7$ and $0.4$ respectively).

Table 1: Datasets used in experiments

| Dataset | #examples | avg. sparsity | #features |
|---|---|---|---|
| 20news | 18845 | 93.89 | 101631 |
| a9a | 48841 | 13.87 | 123 |
| acoustic | 78823 | 50.00 | 50 |
| adult | 48842 | 12.00 | 105 |
| australian | 690 | 11.19 | 14 |
| breast-cancer | 683 | 10.00 | 10 |
| census | 299284 | 32.01 | 401 |
| cod-rna | 271617 | 8.00 | 8 |
| covtype | 581011 | 11.88 | 54 |
| diabetes | 768 | 7.01 | 8 |
| gisette | 1000 | 4971.00 | 5000 |
| heart | 270 | 9.76 | 13 |
| ijcnn1 | 91701 | 13.00 | 22 |
| ionosphere | 351 | 30.06 | 34 |
| letter | 20000 | 15.58 | 16 |
| magic04 | 19020 | 9.99 | 10 |
| mnist | 11791 | 142.43 | 780 |
| mushrooms | 8124 | 21.00 | 112 |
| rcv1 | 781265 | 75.72 | 43001 |
| real-sim | 72309 | 51.30 | 20958 |
| splice | 1000 | 60.00 | 60 |
| w1a | 2477 | 11.47 | 300 |
| w8a | 49749 | 11.65 | 300 |

Table 2: Error rates for Sketched Online Newton with different sketching algorithms

| Dataset | RP-SON | FD-SON | Oja-SON |
|---|---|---|---|
| australian | **15.6** | 16.0 | 15.8 |
| breast-cancer | 4.8 | 5.3 | **3.7** |
| diabetes | 35.5 | 35.4 | **32.8** |
| mushrooms | 0.5 | 0.5 | **0.2** |
| splice | 22.9 | **22.6** | 22.9 |

Table 3: Error rates for different algorithms (with best results bolded)

| Dataset | Oja-SON | | | | ADAGRAD |
|---|---|---|---|---|---|
| | Without Diagonal Adaptation | | With Diagonal Adaptation | | |
| | $m = 0$ | $m = 10$ | $m = 0$ | $m = 10$ | |
| 20news | 0.121338 | 0.121338 | **0.049590** | **0.049590** | 0.068020 |
| a9a | 0.204447 | 0.195203 | **0.155953** | **0.155953** | 0.156414 |
| acoustic | 0.305824 | 0.260241 | **0.257894** | **0.257894** | 0.259493 |
| adult | 0.199763 | 0.199803 | **0.150830** | **0.150830** | 0.181582 |
| australian | 0.366667 | 0.366667 | 0.162319 | **0.157971** | 0.289855 |
| breast-cancer | 0.374817 | 0.374817 | **0.036603** | **0.036603** | 0.358712 |
| census | 0.093610 | 0.062038 | 0.051479 | **0.051439** | 0.069629 |
| cod-rna | 0.175107 | 0.175107 | 0.049710 | **0.049643** | 0.081066 |
| covtype | **0.042304** | **0.042312** | 0.050827 | 0.050818 | 0.045507 |
| diabetes | 0.433594 | 0.433594 | 0.329427 | **0.328125** | 0.391927 |
| gisette | 0.208000 | 0.208000 | **0.152000** | **0.152000** | 0.154000 |
| heart | 0.477778 | 0.388889 | **0.244444** | **0.244444** | 0.362963 |
| ijcnn1 | 0.046826 | 0.046826 | **0.034536** | 0.034645 | 0.036913 |
| ionosphere | 0.188034 | **0.148148** | 0.182336 | 0.182336 | 0.190883 |
| letter | 0.306650 | 0.232300 | 0.233250 | **0.230450** | 0.237350 |
| magic04 | 0.000263 | 0.000263 | **0.000158** | **0.000158** | 0.000210 |
| mnist | 0.062336 | 0.062336 | 0.040031 | **0.039182** | 0.046561 |
| mushrooms | 0.003323 | 0.002339 | 0.002462 | 0.002462 | **0.001969** |
| rcv1 | 0.055976 | 0.052694 | 0.052764 | 0.052766 | **0.050938** |
| real-sim | 0.045140 | 0.043577 | **0.029498** | **0.029498** | 0.031670 |
| splice | 0.343000 | 0.343000 | 0.294000 | **0.229000** | 0.301000 |
| w1a | 0.001615 | 0.001615 | 0.004845 | 0.004845 | **0.003633** |
| w8a | **0.000101** | **0.000101** | 0.000422 | 0.000422 | 0.000221 |

Table 4: Largest relative error between true and estimated top 10 eigenvalues using Oja's rule.

| Dataset | Relative eigenvalue difference |
|---|---|
| a9a | 0.90 |
| australian | 0.85 |
| breast-cancer | 5.38 |
| diabetes | 5.13 |
| heart | 4.36 |
| ijcnn1 | 0.57 |
| magic04 | 11.48 |
| mushrooms | 0.91 |
| splice | 8.23 |
| w8a | 0.95 |

## Footnotes

[7] By adding a suitable constant, these losses can always be made nonnegative while leaving the regret unchanged.

[8] See Appendix B for the closed form of the projection step.