[Reviews · NeurIPS 2016]

Reviewer 1

Summary

The authors strive to make second-order online learning efficient by approximating the scaling matrix A_t by a low-rank sketched version based on S_t^T S_t. They prove scale-invariant regret guarantees for this approach when the desired matrix A_t is well approximated in this way, and show that the algorithm can be implemented efficiently for sparse data.

Qualitative Assessment

The quadratic update time and space requirements of second-order online methods like Online Newton Step make such algorithms unsuitable for most practical problems. The present work takes a significant step in addressing this. The primary contribution of the paper are variations of Online Newton Step that remove this drawback using a sketching approximation to the scaling matrix and a clever implementation of sparse updates. The primary theoretical contributions are the analysis of the RP and FD versions of the algorithm. For RP they show a regret bound which holds when the matrix G_T (the matrix of observed gradients) is actually low-rank. Given the structure of the loss functions assumed, f_t(w) = \ell(< w, x_t >), gradients will always be in the direction of the examples x_t, and so I think this theorem only holds when the data is actually low-rank. But if that was the case you could always simply project the data onto a basis and run ONS on the lower dimensional space. Hence, this result feels somewhat limited. The FD result Thm 3 is thus stronger (and is the one presented in detail in the body of the paper) since it depends instead on the spectral decay of G_T^T G_T. This point should be clarified in the paper. The authors emphasize the fact their results are scale-invariant, but this also comes at some cost. Most results in online convex optimization apply to arbitrary convex functions, possibly satisfying additional conditions like strong convexity or smoothness. This work assumes a much more restrictive class of functions where f_t(w) = \ell(< w, x_t >), essentially a generalized linear model. This appears necessary to introduce the concept of "invariant learning", but the importance of this approach in practice isn't clear to me. One can choose the fixed norm bound on the feasible set after scaling the features, and in practice one can often avoid projecting onto a bounded feasible set at all as long as the learning rate is set reasonably. Further, while the author's claim the bound in Theorem 1 is "qualitatively similar" to the standard setting, the explicit dependence of the dimension d is a significant difference in my mind. The experimental section presents results on both synthetic and real-world datasets. The synthetic data nicely highlights the potential for improvements from this approach; only some of this is demonstrated on real-world datasets. The author's only present experiments on the Oja-SON variant, which is unfortunate since it lacks theoretical guarantees. The importance of the theory is somewhat called into question by this, since it implies the assumptions on A_t necessary for RP-SON and FD-SON to work well may not actually hold in practice (as discussed above, this seems quite likely for RP-SON). Further, in order to outperform AdaGrad (which is 10x faster and substantially easier to implement), a significant trick was needed (lines 254-259). Given that this approach was necessary to get good performance in practice, a more thorough discussion is warranted.

Confidence in this Review

2-Confident (read it all; understood it all reasonably well)


Reviewer 2

Summary

Despite the attractive properties of second-order online methods (e.g., Online Newton Step (ONS)) such as being invariant to linear transformations of the data, but these family of algorithms have a quadratic memory and time dependency to the number of dimensions which limits their practical applicability. This paper aims to improve second-order online methods by integrating random projection and sketching methods into the updating and proposes Sketched version of ONS algorithm ((though with different projections). In particular, this paper achieve regret guarantees nearly as good as the standard regret bounds for ONS, while keeping computation as good as first order methods. The authors prove scale-invariant regret guarantees for their approach and introduce nice tricks for practical implementation of their algorithm which enjoys a running time linear in the sparsity of the examples.

Qualitative Assessment

The problem being studied is interesting and the solution proposed in this paper bridges the gap between nice theoretical properties of ONS and its practical value. The presentation of the paper was mostly clear. The claimed contributions are discussed in the light of existing results and the paper does survey related work appropriately. The paper is technically sound and the proofs seem to be correct as far as I checked. From a theoretical standpoint, the paper presents regret analysis for ONS when Random Projection (RP) and Frequent Directions (FD) are used to sketch the matrix G_T (the T by d matrix of sequence of T gradients). The results holds when the sketching dimension m is roughly \Omega (r + log T), where r is assumed to be rank of the G_T which is equivalent to rank of data in their setting. This means when data points lie in a low-rank manifold, then ONS with random projection can be utilized to improve the running time. I think the statement of theorems needs to be stated clearly at least as a Remark in terms of rank of actual data points rather than the rank of G_T. Empirically, three sketching approaches as well as a sparse implementation are evaluated on both synthetic and real world datasets. While the experiments on synthetic data looks promising, but I think the experiments on real-world datasets in the current status does not fully complement the theoretical achievements of the paper and needs to be strengthened (or at least needs through discussion to convince the reader). First of all, my first impression from experiments is that the neither the RP-SON nor FD-SON which come with strong theoretical guarantees can outperform Oja-SON which unfortunately does not come with theoretical analysis. Also, the Oja-SON was outperformed by AdaGrad, however, using a simple diagonal re-scaling results in much better results which need through discussions. Overall I liked the idea of sketching in second order online optimization and its analysis, and I lean towards acceptance (assuming the paper will honestly discuss the issues mentioned earlier).

Confidence in this Review

2-Confident (read it all; understood it all reasonably well)


Reviewer 3

Summary

This paper proposed an improved ONS method with better regret guarantees and computation cost. The overall contribution contains three part: 1. relax the assumption about fixed norm to bounded predictions. 2. tackle 3 matrix sketching techniques to reduce the computation cost and prove regret bounds which is similar with the full second order update. 3. Develop sparse versions of these updates with a running time linear in the sparsity of the examples . Some empirical analysis has been performed to demonstrate superiority of the proposed method.

Qualitative Assessment

Specifically, this work first relaxed the assumption about fixed norm in ONS method and prove a similar regret bound, then replaced the full matrix update with sketching matrix (RP-sketch, FD-sketch and Oja’s sketch) to reduce the computation cost, they also provide similar regret bound. Finally, they proposed sparse version of these 3 algorithms. Quilaty: From a theoretical perspective, this paper is really good. They provide a similar logarithmic regret bound with relaxed assumption. To reduce the computation cost about the second order information, they adopt 3 matrix sketching method to approximate the full second order matrix and prove similar regret bound. This analysis can be generalized to other matrix sketching method. The weakest point of this paper is the experiments. In my opinion, full matrix update can get a little better improvement than diagonal matrix update but take much more time in many real-world datasets. This paper claimed the effectiveness of employing matrix sketching techniques, but there is no experiments about computation cost analysis. Clarity: This paper is easy to follow. Significance: Adagrad is widely used in different machine learning community. This paper is a step toward replacing the full matrix update with sketched matrix update to reduce the computation cost. Other comments: 1 I am confused why the authors proposed a lower bound in theorem 1. 2 line 341 in appendix, can you give more details about this inequality that uses Khintchine inequlity. 3 line 373 in appendix, Assumption 1 -> Assumption 2 Should \|w\|^2_{A_0} be \|w – w_1 \|^2_{A_0} ?

Confidence in this Review

2-Confident (read it all; understood it all reasonably well)


Reviewer 4

Summary

This paper introduces a sketched version of the Online Newton algorithm which enjoys runtime respectively linear, O(md), in the data dimension (d) and in the sketch size (m). The proposed algorithm (SON) enjoys improved regret gaurantees (bounded predictions instead of solutions) for ill-conditioned matrices. Three sketching approaches as well as a sparse implementation are defined and tested on both synthetic and real world datasets. Experiments with an AdaGrad-flavored SON show strong empirical performance although this algorithm is unfounded theoretically. A truly condition-invariant algorithm which uses the Moore-Penrose pseudoinverse as opposed to the true inverse is defined and proven in the Appendix.

Qualitative Assessment

Using sketching, the authors provide what looks like the first linear-time, second-order, online learning algorithm with an invariant regret guarantee. They also provide a sparse implementation with these properties along with 3 different sketching techniques. This work significantly improves upon Online Newton Step in a non-trivial way. The strength of this paper is in the algorithm design and proofs, however some emiprical results are exemplary; the Oja version of SON was outperformed by AdaGrad, however, incorporating a simple modification gave superior results. Overall, SON is the first to reach the linear time benchmark for second order online learning which merits recognition by itself. The sparse implementation, sketching variants, and AdaGrad-flavored Oja-SON put the paper above and beyond expectations. Comments: 1) Comparison to Online Frank-Wolfe would be appreciated

Confidence in this Review

2-Confident (read it all; understood it all reasonably well)


Reviewer 5

Summary

This paper considers an online algorithm similar to online newton step proposed by (Hazan et al. 2007). The proposed algorithm is invariant under linear transformation of the data. The definition of regret in this paper differs for the usual definition. Instead of requiring that predictor vector belong to a fixed compact convex set, they require the output of the linear predictor applied to each data point be in a bounded interval [-C, C]. This definition is taken from (Ross et al. 2013) which also consider scale invariant online algorithms. They provide optimal regret bounds under different assumptions on the loss functions. To reduce the per-iteration cost of their algorithm and required memory for storage of the A_t matrix in their algorithm, the authors consider three sketching methods that maintain a low-rank approximation of that matrix. They provide regret bounds for two of the three sketching methods that depend on the rank in the low-rank approximation of A_t. They also show how to implement these algorithms such that the run-time depend on the sparsity of example data vectors rather than their dimension.

Qualitative Assessment

According to the numerical experiments section, the third sketching algorithm has the best performance. However, this method does not have a regret bound. The first two sketching methods with regret bounds have been skipped over in the numerical experiments. This reviewer believes that the experiments section should include the first two sketching algorithms since these are the ones for which regret bound guarantees are presented in the paper. The authors have not commented on how much using a different definition of regret makes the proof of regret bounds different for their algorithm. Since for the classical definition of regret the online newton step has been analyzed for exp-concave loss functions, it is important to explicitly compare the proof of theorem 1 in their paper to the regret bound proofs in the literature (e.g. in Hazan et al. 2007). For example, a question that should be answered is: Can the proof be extended for any convex compact set K_t in the definition of regret?

Confidence in this Review

2-Confident (read it all; understood it all reasonably well)


Reviewer 6

Summary

This paper studies sketching techniques in the context of online Newton and applies Oja's algorithm to online Newton.

Qualitative Assessment

The paper has two different components. One is an extensive theoretical analysis of random projections and forward directions applied to the online Newton, with regret bounds, a description of their sparse implementations and their running times. However, the authors do not show any sort of experimental results for either of these two sketching methods. On the other thand, the authors implemented Oja's sketch method for online Newton and compared it to AdaGrad. There is an improvement in a handful of the datasets tested. Unlike the previous two methods, this method comes with little theoretical justification. These two parts are not joined very well, and feels like a 'theory half' and a 'experimental half' that are only related by the topic of sketching in online learning and not in the specifics. It would be much more cohesive if, say, the authors included experimental results for the first two methods (FD and RP) even if they were not competitive, for comparison's sake. Especially since the sparse implementations of the theoretically justified methods are covered in such detail. Minor comment: Table 2 of the appendix should have the best result in each row bolded or emphasized. The equation after line 409 in the appendix should be an inequality in the second line.

Confidence in this Review

2-Confident (read it all; understood it all reasonably well)